# Laminar shear stress inhibits inflammation by activating autophagy in human aortic endothelial cells through HMGB1 nuclear translocation

Qingyu Meng[1], Luya Pu[1], Mingran Qi[1], Shuai Li[1], Banghao Sun[1], Yaru Wang[1], Bin Liu [2✉] & Fan Li [1,3,4,5,6✉]

Prevention and treatment of atherosclerosis (AS) by targeting the inflammatory response in vascular endothelial cells has attracted much attention in recent years. Laminar shear stress (LSS) has well-recognized anti-AS properties, however, the exact molecular mechanism remains unclear. In this study, we found that LSS could inhibit the increased expression of intercellular adhesion molecule-1 (ICAM-1), vascular cell adhesion molecule-1 (VCAM-1), cyclooxygenase-2 (COX-2), and matrix metallopeptidase-9 (MMP-9) caused by TNF-α in an autophagy-dependent pathway in human aortic endothelial cells (HAECs) and human umbilical vein endothelial cells (HUVECs). Whole-transcriptome sequencing analysis revealed that erythropoietin-producing hepatocyte receptor B2 (*EPHB2*) was a key gene in response to LSS. Moreover, co-immunoprecipitation assay indicated that LSS could enhance the EPHB2-mediated nuclear translocation of high mobility group box-1 (HMGB1), which interacts with Beclin-1 (BECN1) and finally leads to autophagy. Simultaneously, we identified an LSS-sensitive long non-coding RNA (lncRNA), LOC10798635, and constructed an LSS-related LOC107986345/miR-128-3p/EPHB2 regulatory axis. Further research revealed the anti-inflammatory effect of LSS depends on autophagy activation resulting from the nuclear translocation of HMGB1 via the LOC107986345/miR-128-3p/EPHB2 axis. Our study demonstrates that LSS could regulate the expression of EPHB2 in HAECs, and the LOC107986345/miR-128-3p/EPHB2 axis plays a vital role in AS development.

[1] Department of Pathogenobiology, The Key Laboratory of Zoonosis, Chinese Ministry of Education, College of Basic Medicine, Jilin University, Changchun, China. [2] Cardiovascular Disease Center, The First Hospital of Jilin University, Changchun, China. [3] Engineering Research Center for Medical Biomaterials of Jilin Province, Jilin University, Changchun, China. [4] Key Laboratory for Health Biomedical Materials of Jilin Province, Jilin University, Changchun, China. [5] State Key Laboratory of Pathogenesis, Prevention and Treatment of High Incidence Diseases in Central Asia, Xinjiang, China. [6] The Key Laboratory for Bionics Engineering, Ministry of Education, Jilin University, Changchun, China. ✉email: binliu@jlu.edu.cn; lifan@jlu.edu.cn

With rapid development of the social economy and population aging, cardiovascular and cerebrovascular diseases caused by AS have become one of the most important diseases that seriously threaten human life and health[1]. AS development roughly encompasses the following processes: endothelial cell inflammatory injury and monocyte adhesion; foam cell formation; smooth muscle cell migration and proliferation from media to intima; and atherosclerotic plaque formation[2]. Inflammatory cytokines and other acute reactants released by endothelial cells during inflammation accelerate endothelial dysfunction[3]. Therefore, an in-depth study of the pathogenesis of AS from the perspective of inflammation is of great significance for the prevention and treatment of this disease.

LSS is the frictional force of blood flow acting on the per unit surface of vascular endothelial cells. It is essential for regulating the occurrence and development of AS and the normal physiological functions of blood vessels[4]. Studies have shown that LSS plays an important role in anti-inflammation, anti-adhesion, anti-aging, and protection of endothelial cells by affecting gene expression in vascular endothelial cells, thereby delaying the occurrence and development of AS[5].

It has been proven that autophagy is involved in a variety of biological processes in cells[6,7], especially in the context of endothelial cell inflammation[8,9]. Studies have shown that macrophages play a key role in destroying AS plaques[10,11]. Therefore, autophagy-targeted treatment of AS has important clinical significance. The relationship between LSS and endothelial cell autophagy has been studied extensively[12–14]. Research has shown that functional hyperemia caused by rhythmic grip exercise can increase the arterial shear rate and activate autophagy in primary arterial endothelial cells of healthy males[15]. HAECs and bovine aortic endothelial cells exposed to LSS also show a higher degree of intracellular autophagy, indicating that autophagy may be the key executor of the protective effect of LSS in AS[16,17].

HMGB1 is a ubiquitous nuclear protein that binds to DNA, which contributes to the stability of chromatin structure and the regulation of target gene transcription[18]. When cells are activated or damaged, HMGB1 will translocate to the cytoplasm or extracellular space, cytoplasmic HMGB1 can enhance autophagy in response to cellular stress. After cytoplasmic HMGB1 replaces Bcl-2 of BECN1, it directly binds to BECN1 to induce autophagosome formation[19,20]. Therefore, HMGB1 is considered to be a key regulator of autophagy. EPHB2 is also widely involved in the study of autophagy. Zhong et al. showed that the EPHB2-mediated forward signaling pathway could activate autophagy to alleviate Aβ-induced endoplasmic reticulum stress and apoptosis in HT22 cells[21]. A recent study found that *RNF186* encodes an E3 ubiquitin-protein ligase that regulates ubiquitination of EPHB2 and further recruits microtubule-associated protein 1 light chain 3 beta (MAP1LC3B) for autophagy, which ultimately relieves dextran sodium sulfate-induced mouse colitis[22]. However, the relationship of HMGB1 and EPHB2 is still poorly characterized to date.

A large number of experiments have shown that some of the genes in vascular endothelial cells are LSS-sensitive genes[23,24], which precisely regulates cell activity. In recent years, long noncoding RNAs (lncRNAs) have become an important "regulator" in the gene expression network[25–27]. An increasing amount of evidence shows that lncRNAs can be used as competitive endogenous RNAs (ceRNAs) to affect various biological processes, playing a key role in the development of cardiovascular disease[28,29]. One of our previous studies showed that the lncRNA, AF131217.1, plays an anti-inflammatory role by competitively binding to miR-128-3p to regulate the expression of Kruppel-like factor 4 in HUVECs[30]. However, the expression and function of lncRNAs in HAECs under LSS remain largely unclear.

In this study, we used whole-transcriptome sequencing technology to analyze the expression of EPHB2 after LSS acted on HAECs, and verified the interaction between EPHB2 and HMGB1 under the action of LSS. At the same time, we identified a novel LSS-sensitive lncRNA, LOC10798635, and constructed a LOC107986345/miR-128-3p/EPHB2 regulatory axis, further reveals the molecular mechanism of LSS inhibiting endothelial cell inflammation by activating autophagy.

## Results

**LSS inhibits the TNF-α-induced inflammatory response by activating autophagy in endothelial cells.** To explore the relationship between LSS, autophagy, and endothelial cell inflammation, we selected HAECs and HUVECs as in-vitro models. First, both cell types were treated with 12 dyn/cm² LSS for 12 h, and then LSS was removed to return the cells to stable culture for 24 h. The results showed that the expression of the autophagy marker protein, MAP1LC3B2, was increased, and the expression of sequestosome-1/P62 (SQSTM1/P62) was decreased after the action of LSS on endothelial cells, indicating that LSS significantly activates autophagy. However, autophagy activation disappeared after withdrawal of LSS, indicating that LSS is closely related to endothelial autophagy (Fig. 1a–d). To investigate whether the observed accumulation of MAP1LC3B2 was due to LSS-induced activation of autophagy or an inhibition of the lysosomal degradation pathway, we co-treated HAECs and HUVECs with LSS and chloroquine (CQ). As expected, both treatments increase the amount of MAP1LC3B2 level, and notably, co-treatment with LSS and CQ induced significantly more MAP1LC3B2 accumulation than CQ treatment alone (Fig. 1e–h). These results indicate that the observed accumulation of MAP1LC3B2 is due to LSS-induced activation of autophagy rather than inhibition of the lysosomal degradation pathway. Then, we used TNF-α to stimulate endothelial cell inflammation to observe the anti-inflammatory effect of LSS. The results show that TNF-α significantly upregulates the expression of ICAM-1, VCAM-1, COX-2, and MMP-9 in HAECs and HUVECs. However, LSS inhibited TNF-α-induced inflammatory factor concentrations in HAECs and HUVECs, and at the same time, autophagy was significantly activated (Fig. 1i–l). Further research found that when the autophagy pathway was inhibited by 3-Methyladenine (3-MA) (Fig. 1m–p) or BECN1 small interfering RNA (BECN1 siRNA) (Fig. S1a-d), the anti-inflammatory effect of LSS disappeared, indicating that LSS could inhibit the inflammatory response in endothelial cells by activating autophagy.

**Whole-transcriptome sequencing analysis reveals that *EPHB2* was a key gene in response to LSS.** To observe the changes in gene expression in endothelial cells before and after application of LSS, we performed whole-transcriptome sequencing on the three groups of HAECs (Static, LSS, and After LSS). Principal component analysis (PCA) was performed on the messenger RNA (mRNA) expression profile obtained by sequencing. The differences within the three groups of samples were small, whereas the between-group differences were large, indicating that the quality of sequencing results was high (Fig. 2a). Subsequently, Short Time-series Expression Miner was used to search for genes with the same expression pattern. In Module 10, the expression of this part of mRNAs showed an increasing trend at first, followed by a decrease (Fig. 2b). To screen LSS-sensitive genes more accurately, we used the DESeq2 package of R to analyze sequencing data. We took the intersection of upregulated genes after LSS, down-regulated genes after restoration of steady-state culture, and genes for which expression did not change in Static and After LSS group, finally got 96 differentially expressed (DE) mRNAs (Fig. 2c, Supplementary Data 2). We then used the pheatmap

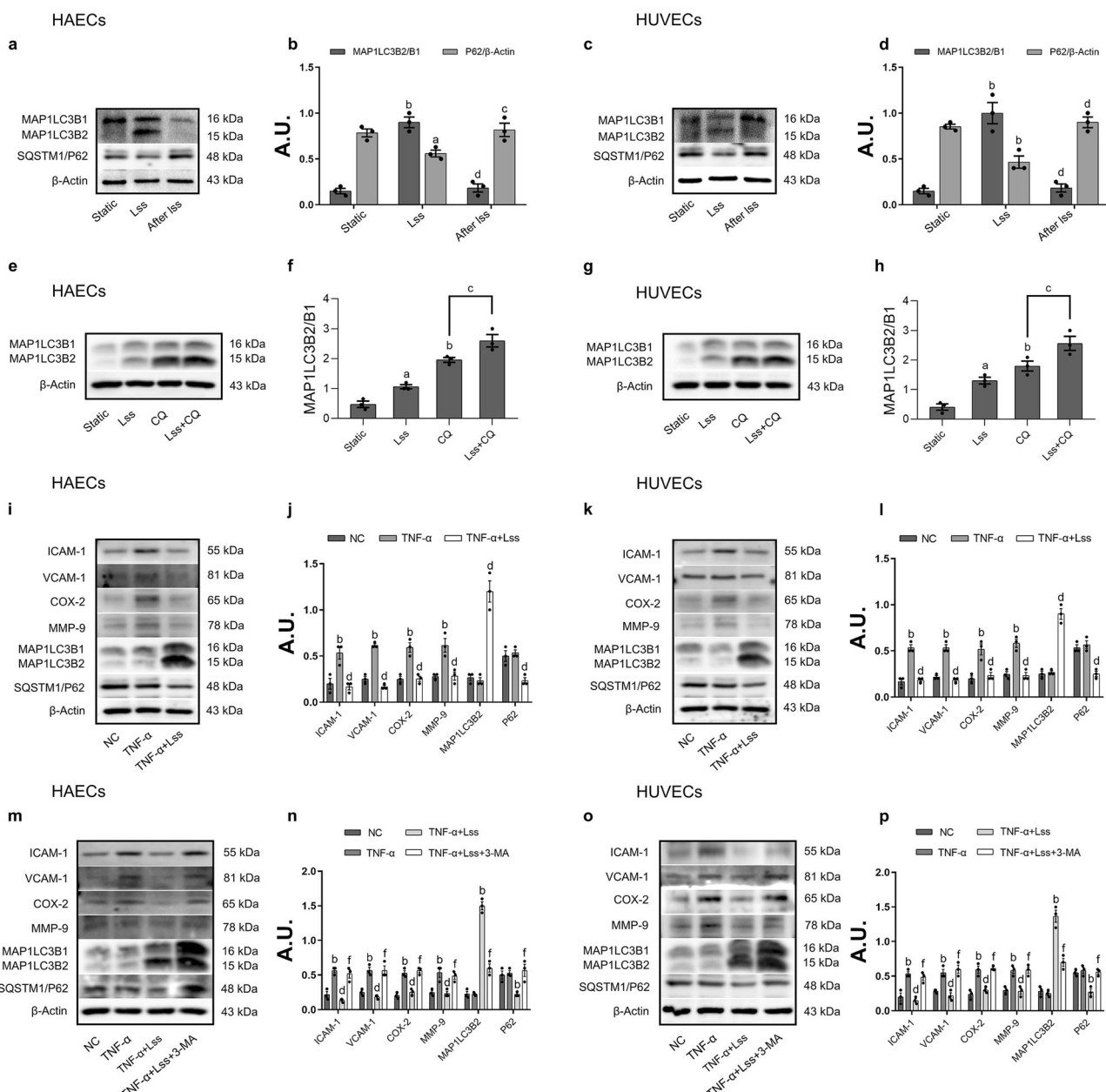

**Fig. 1 LSS inhibits the inflammatory response in endothelial cells by activating autophagy. a**, **c** Western Blot shows the expression of MAP1LC3B2 and SQSTM1/P62 in HAECs and HUVECs. **b**, **d** Relative expression of MAP1LC3B2 and SQSTM1/P62 in HAECs and HUVECs. Cells were treated with 12 dyn/cm$^2$ LSS for 12 h, CQ (50 μM) precedes LSS for 6 h, and **e**, **g** Western Blot shows the expression of MAP1LC3B2 in HAECs and HUVECs. **f**, **h** Relative expression of MAP1LC3B2 in HAECs and HUVECs. Cells were stimulated with TNF-α (10 ng/ml) for 12 h and then treated with LSS for another 12 h, and **i**, **k** Western Blot shows the expression of ICAM-1, VCAM-1, COX-2, MMP-9, MAP1LC3B2, and SQSTM1/P62 in HAECs and HUVECs. **j**, **l** Relative expression of ICAM-1, VCAM-1, COX-2, MMP-9, MAP1LC3B2, and SQSTM1/P62 in HAECs and HUVECs. Cells were pretreated with TNF-α (10 ng/ml) for 12 h and in the presence or absence of 3-MA (5 mM) for another 2 h, followed by LSS treatment for another 12 h, and **m**, **o** Western Blot shows the expression of ICAM-1, VCAM-1, COX-2, MMP-9, MAP1LC3B2, and SQSTM1/P62 in HAECs and HUVECs. **n**, **p**. Relative expression of ICAM-1, VCAM-1, COX-2, MMP-9, MAP1LC3B2, and SQSTM1/P62 in HAECs and HUVECs. Data are presented as mean ± SEM of three independent experiments. $^{a}P < 0.05$, $^{b}P < 0.01$ vs. Static or NC group; $^{c}P < 0.05$, $^{d}P < 0.01$ vs. LSS or TNF-α or CQ group; $^{e}P < 0.05$, $^{f}P < 0.01$ vs. TNF-α + LSS group. AU arbitrary units.

package of R to draw a cluster analysis heat map of DE mRNAs. The results show that LSS can significantly change the expression profile of mRNAs (Fig. 2d) in HAECs. We used R to perform Gene Ontology (GO), Kyoto Encyclopedia of Genes and Genomes (KEGG), and disease correlation analyses on 96 DE mRNAs. The GO analysis shows that the main biological process of DE mRNA enrichment is the cellular response to extracellular stimulation (Fig. 2e). In the KEGG pathway analysis, DE mRNAs were mainly involved in cell signaling pathways, such as the

pentose phosphate and mTOR signaling pathway and cytokine–cytokine receptor interactions, amongst others (Fig. 2f). The results of the disease correlation analysis show that DE mRNAs may be related to cardiovascular diseases, such as congestive heart failure and retinal vascular disease (Fig. 2g). The protein-protein interaction (PPI) network analysis of 96 DE mRNAs shows that EPHB2 had a higher binding score (Fig. 2h), and EPHB2 was closely related to autophagy and AS[31]. Therefore, *EPHB2* was selected as the core regulatory gene, and Cytoscape

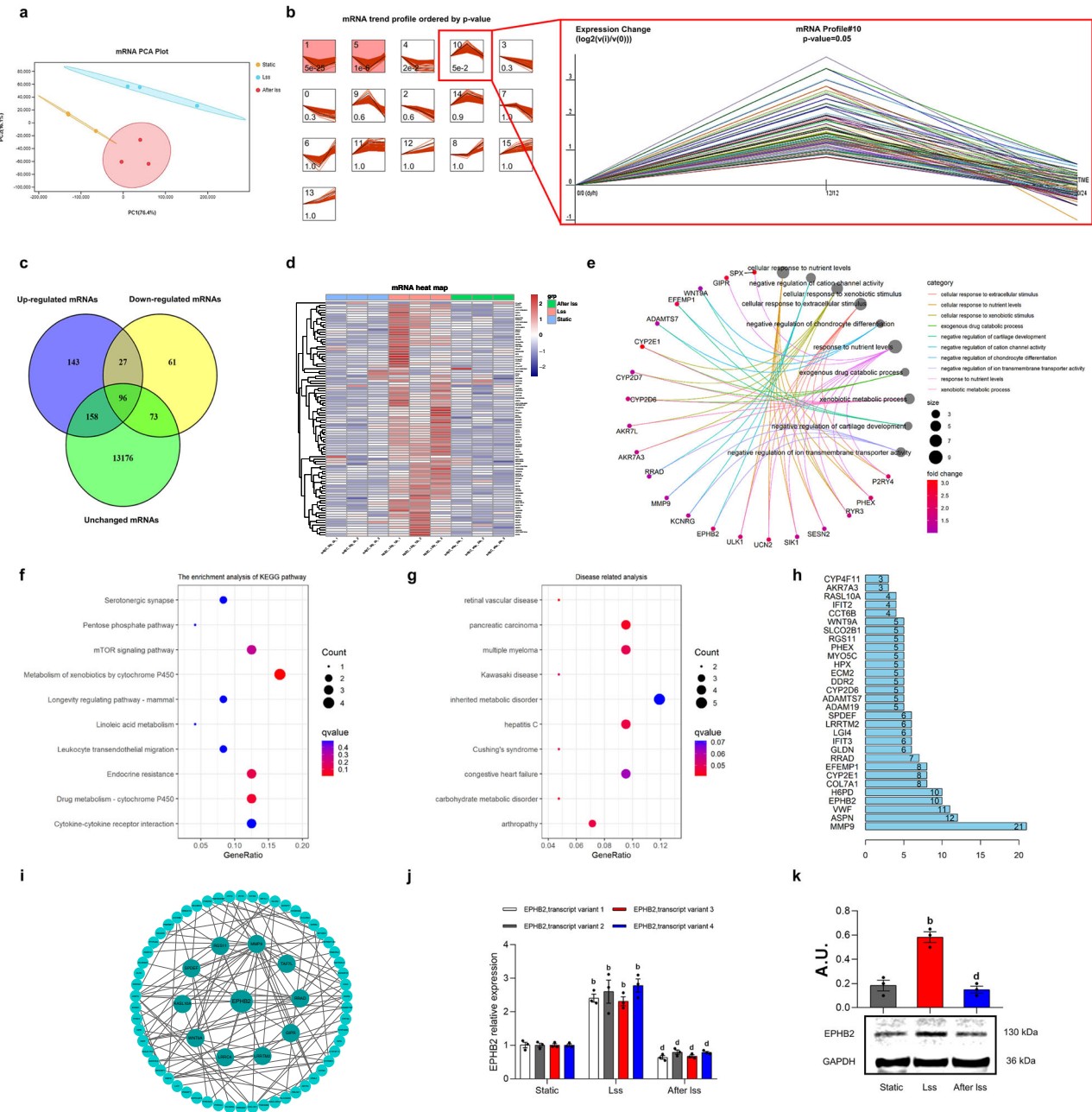

**Fig. 2 Whole-transcriptome sequencing analysis revealed that EPHB2 was a key gene in response to LSS. a** mRNA PCA analysis of HAECs in three groups. **b** mRNA trend profiles ordered by *P* value. Expression Change (log2(v(i)/v(0))). **c** DE mRNA Venn diagram. **d** DE mRNA heat map. **e** DE mRNA GO biological process analysis. **f** DE mRNA KEGG pathway analysis. **g** DE mRNA disease correlation analysis. **h** PPI histogram of DE mRNA. **i** PPI network diagram with EPHB2 as the core. **j** qRT-PCR shows the expression of EPHB2. **k** Western Blot shows the expression of EPHB2. Data are presented as mean ± SEM of three independent experiments. [a]*P* < 0.05, [b]*P* < 0.01 vs. Static group; [c]*P* < 0.05, [d]*P* < 0.01 vs. LSS group. AU arbitrary units.

software was used to construct a PPI network diagram with EPHB2 at its core (Fig. 2i). Simultaneously, we detected the expression of EPHB2 in the three groups of endothelial cells. The results of quantitative real-time polymerase chain reaction (qRT-PCR) (Fig. 2j) and Western blot (Fig. 2k) show that EPHB2 expression was significantly increased by LSS and decreased after removal of LSS, revealed that EPHB2 was a key gene in response to LSS in HAECs.

**LSS activates autophagy flux by enhancing the EPHB2-mediated nuclear translocation of HMGB1.** To explore the role of EPHB2 in the process of LSS-induced endothelial cell autophagy, we used the EPHB2 knockdown plasmids (sh-

EPHB2-1, sh-EPHB2-2, and sh-EPHB2-3) to reduce its expression, and the results showed that the plasmid sh-EPHB2-2 can significantly reduce the expression of EPHB2 (Fig. 3a, b). The results of Western blot showed that when the expression of EPHB2 was inhibited by sh-EPHB2-2, the increased expression of MAP1LC3B2 and the decreased expression of P62 disappeared after LSS acted on HAECs (Fig. 3c). Later, acidic lysosomes were stained with Lyso Tracker Red (LTR) fluorescent dye to observe the autophagy induction of LSS after EPHB2 was knockdown, the results show that inhibiting the expression of EPHB2 can significantly reduce the red fluorescence intensity caused by LSS (Fig. 3d, e). Finally, we infected HAECs with mRFP-GFP-LC3 adenovirus to monitor autophagy flux. Our results showed that in

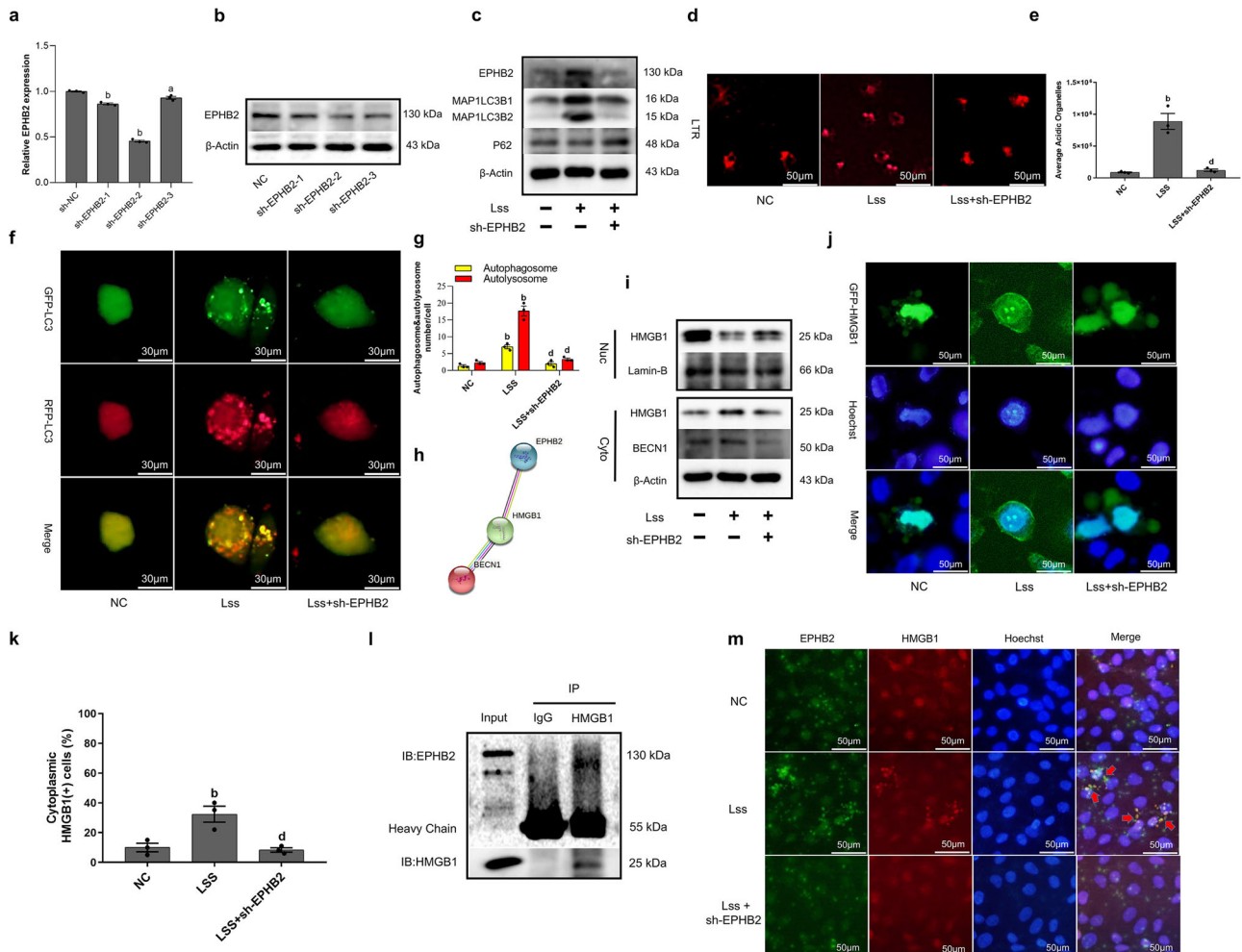

**Fig. 3 LSS activates autophagy flux by enhancing the EPHB2-mediated nuclear translocation of HMGB1. a** qRT-PCR shows the expression of EPHB2. **b** Western Blot shows the expression of EPHB2. **c** Western Blot shows the expression of EPHB2, MAP1LC3B2, and SQSTM1/P62. **d, e** Confocal microscopy shows HAECs stained with Lyso Tracker Red, and lysosomes show red fluorescence. **f, g** The mRFP-GFP-LC3 adenovirus were used to observe autophagic flux under a laser confocal microscope. mRFP was used to label and track LC3. Weakening of GFP indicates fusion of lysosomes and autophagosomes to form autophagolysosomes. **h** PPI network functional enrichment analysis of EPHB2, HMGB1, and BECN1. **i** The nucleoplasmic separation experiment shows the subcellular localization of HMGB1. **j, k**. GFP-HMGB1 plasmid were used to observe the nuclear translocation of HMGB1 under a laser confocal microscope. **l** Co-immunoprecipitation assay shows the physical interaction between HMGB1 and EPHB2 in HAECs under the action of LSS, and the cell lysates were immunoprecipitated by anti-HMGB1 antibody. m. Immunofluorescence assay shows the co-localization of EPHB2 and HMGB1 in HAECs under the action of LSS. Data are presented as mean ± SEM of three independent experiments. $^{a}P < 0.05$, $^{b}P < 0.01$ vs. NC group; $^{c}P < 0.05$, $^{d}P < 0.01$ vs. LSS group.

the presence of the plasmid of sh-EPHB2-2, the autophagy flow induced by LSS was significantly inhibited (Fig. 3f, g). The above results indicated that LSS can activate endothelial cell autophagy through EPHB2. Next, we want to explore whether the ability that LSS activates endothelial cell autophagy through EPHB2 was related to the nuclear translocation of HMGB1. First, the PPI network analysis of EPHB2, HMGB1, and BECN1 showed that there was an interaction between the three of them (Fig. 3h), then we extracted the nucleoplasmic protein of HAECs, the results showed that HMGB1 was transported out of the nucleus under the action of LSS, and the phenomenon disappeared after the expression of EPHB2 was suppressed (Fig. 3i). A consistent phenomenon was also observed when transfected the plasmid expressing GFP-HMGB1 into HAECs (Fig. 3j, k). In order to explore the relationship between EPHB2 and HMGB1 in HAECs stimulated by LSS, co-immunoprecipitation assay was used to observe whether they can interact, the results showed that there was an interaction between EPHB2 and HMGB1 (Fig. 3l).

Subsequent immunofluorescence experiments proved that HMGB1 can exit the nucleus and co-localize with EPHB2 under the action of LSS (Fig. 3m).

**Overexpression of EPHB2 can activate autophagy and exert the same anti-inflammatory effect as LSS.** As we expected, LSS can activate autophagy to inhibit the expression of inflammatory factors while enhancing the EPHB2-mediated nuclear translocation of HMGB1, when EPHB2 was inhibited by its knockdown plasmid sh-EPHB2-2, the anti-inflammatory effect of LSS was greatly weaken accompanied by a decrease in autophagy levels (Fig. S1e–g). Correspondingly, when we inhibited the expression of HMGB1 with its inhibitor glycyrrhizin acid (GA), the anti-inflammatory effect of LSS was also greatly weakened accompanied by a decrease in autophagy levels (Fig. 4a, b). LTR and monocyte adhesion experiments further proved our conclusions (Fig. 4c). Next, we want to verify whether overexpression of EPHB2 can exert the same anti-inflammatory effect as LSS by

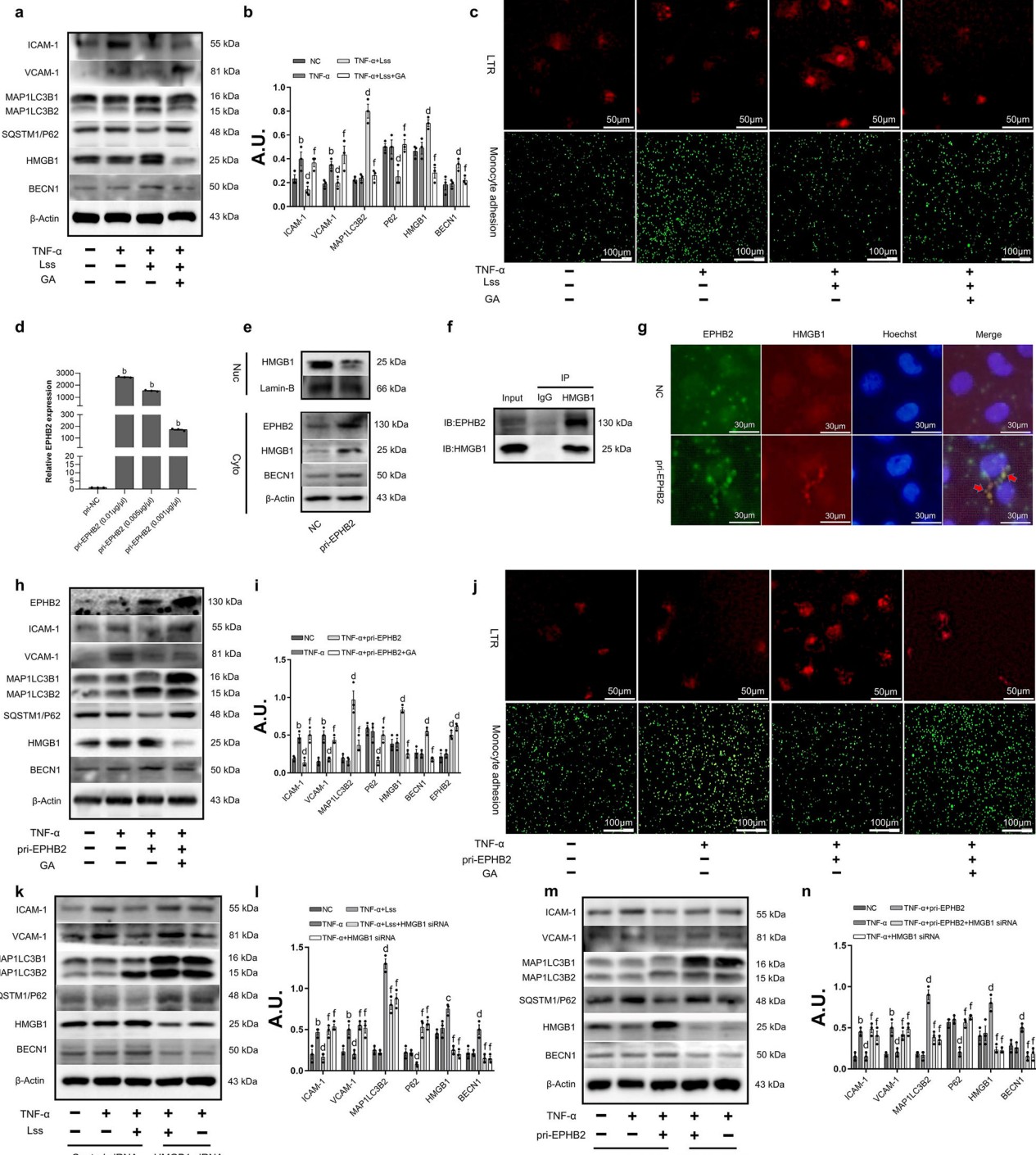

**Fig. 4 Overexpression of EPHB2 can activate autophagy and exert the same anti-inflammatory effect as LSS. a**, **b** Western Blot shows the effect of GA (100 μM) on anti-inflammatory effect of LSS. **c** Confocal microscopy shows the effect of GA on LSS activated autophagy (HAECs stained with Lyso Tracker Red) and inhibited monocyte adhesion. **d** qRT-PCR shows the expression of EPHB2. **e** The nucleoplasmic separation experiment shows the subcellular localization of HMGB1 when EPHB2 was overexpressed. **f** Co-immunoprecipitation assay shows the physical interaction between HMGB1 and EPHB2 in HAECs when EPHB2 was overexpressed, and the cell lysates were immunoprecipitated by anti-HMGB1 antibody. **g** Immunofluorescence assay shows the co-localization of EPHB2 and HMGB1 in HAECs when EPHB2 was overexpressed. **h**, **i** Western Blot shows the effect of GA on anti-inflammatory effect of pri-EPHB2. **j** Confocal microscopy shows the effect of GA on pri-EPHB2 activated autophagy (HAECs stained with Lyso Tracker Red) and inhibited monocyte adhesion. **k**, **l**. Western Blot shows the effect of HMGB1 siRNA on anti-inflammatory effect of LSS. **m**, **n**. Western Blot shows the effect of HMGB1 siRNA on anti-inflammatory effect of pri-EPHB2. Data are presented as mean ± SEM of three independent experiments. [a]$P < 0.05$, [b]$P < 0.01$ vs. NC group; [c]$P < 0.05$, [d]$P < 0.01$ vs. TNF-α group; [e]$P < 0.05$, [f]$P < 0.01$ vs. TNF-α + LSS or pri-EPHB2 group. AU, arbitrary units.

inducing the nuclear translocation of HMGB1. Interestingly, we observed that overexpression of EPHB2 could induce the nuclear translocation of HMGB1 (Fig. 4d, e) and then interact with EPHB2 (Fig. 4f, g), and GA can also counteract the anti-inflammatory effects of overexpression of EPHB2 by inhibiting autophagy (Fig. 4h–j). Moreover, the anti-inflammatory effect of LSS and overexpression of EPHB2 were reduced when HAECs were transfected with HMGB1 siRNA, accompanied by a decrease in MAP1LC3B2 expression and an increase in SQSTM1/P62 expression (Fig. 4k–n).

**Construction of the LSS-related LOC107986345/miR-128-3p/ EPHB2 network**. An increasing amount of evidence shows that lncRNAs can be used as ceRNAs to affect various biological processes, playing a key role in the development of cardiovascular disease. We want to explore whether EPHB2 was regulated by certain ceRNAs mechanisms when HAECs were under the action of LSS. The lncRNAs data obtained by whole-transcriptome sequencing were processed in the same way as the mRNAs before, then we obtained 36 DE lncRNAs (Fig. 5a, Supplementary Data 3) and used the pheatmap package of R to draw a cluster analysis heat map of DE lncRNAs (Fig. 5b). Subsequently, 265 micro RNAs (miRNAs) that potentially bind to EPHB2 (Supplementary Data 4) were predicted through the TargetScan, miRDB, and DIANA databases and intersected with the miRNAs predicted by 36 DE lncRNAs (Supplementary Data 5) to obtain 17 key miRNAs (Table S1). We used Cytoscape software to construct a lncRNA/ miRNA/mRNA regulatory network with EPHB2, 17 target miRNAs, and corresponding lncRNAs (Fig. 5c). Subsequently, the 17 target miRNAs were verified by qRT-PCR, and the results show that only the expression of has-miR-128-3p was in line with the expected conjecture (an initial decrease, followed by an increase) (Fig. 5d). The corresponding LOC107986345 (NCBI Reference Sequence: XR_001742403.1, Table S2) and LOC107986196 were verified by qRT-PCR (Fig. 5e). The results showed that the expression of the two lncRNAs was in line with expectations. Subsequently, we constructed the LOC107986345/miR-128-3p/ EPHB2 and LOC107986196/miR-128-3p/EPHB2 regulatory axes (Fig. 5f). Since the subsequent LOC107986196/miR-128-3p/ EPHB2 axis was not successfully verified, the LSS-related LOC107986345/miR-128-3p/EPHB2 regulatory axis was constructed and a flowchart of this part of the research was shown in Fig. S2. Next, we verified the targeting relationship between LOC107986345 and miR-128-3p/EPHB2. First, fluorescence in-situ hybridization (FISH) was executed to identify the localization of LOC107986345 in HAECs, the results indicated that LOC107986345 was located in both the cytoplasm and nucleus whether the HAECs are in static or under the action of LSS (Fig. 5g). Next, we used nuclear and cytoplasmic RNA isolation technology to extract nuclear and cytoplasmic RNA separately and detect the expression of LOC107986345 by qRT-PCR. The results showed that LOC107986345 was mainly distributed in the cytoplasm (Fig. 5h). Coding potential assessment tool (CPAT) was used to predict the coding ability of LOC107986345. The prediction results showed that LOC107986345 does not have a protein-coding ability (Fig. 5i). The distribution of LOC107986345 was predicted by the lncRNA subcellular location prediction website, lncLocator. The results showed that the score in the cytoplasm was 0.88 (Fig. 5j). Then we use the Mfold web server (http://www. unafold.org/mfold/applications/rna-folding-form.php) to predict the secondary structure of LOC107986345 and use the dual-luciferase reporter assays to verify that there was a binding site between LOC107986345 and miR-128-3p, it also proves that EPHB2 has a strong targeting relationship with miR-128-3p (Fig. 5k–m).

To further verify whether miR-128-3p can bind to LOC107986345 and EPHB2, we used RNA-binding protein immunoprecipitation (RIP) technology. The results of RIP showed that miR-128-3p can bind to LOC107986345 and EPHB2 (Fig. 5n–q). qRT-PCR was used to validate the interaction between LOC107986196, miR-128-3p, and EPHB2. The results show that overexpression of LOC107986345 had the most obvious inhibitory effect on miR-128-3p in endothelial cells at 0.001 μg/μL. At the same time, the expression of EPHB2 was also significantly increased (Fig. S3a–c). LOC107986345 knockout plasmid No. 1 (sh-LOC107986345-1) significantly promoted the expression of miR-128-3p, and the expression of EPHB2 also decreased accordingly (Fig. S3d–g). Similarly, the results of overexpression and knockdown of miR-128-3p also proved that LOC107986345, miR-128-3p, and EPHB2 can be mutually regulated (Fig. S3h–o). Next, we co-transfected the overexpressed or knocked down plasmids of LOC107986345 and miR-128-3p into endothelial cells to observe the expression of EPHB2. qRT-PCR and Western blot results show that the co-transfection plasmid partially reversed the decrease in EPHB2 expression caused by overexpression of the miR-128-3p-transfected plasmid alone and the increase in EPHB2 expression caused by knock-down of the miR-128-3p-transfected plasmid alone (Fig. S3p–s). These results indicated that the LSS-related LOC107986345/miR-128-3p/EPHB2 regulatory axis was successfully validated in HAECs.

**The inhibition of endothelial inflammatory by LSS depends on autophagy activation resulting from the nuclear translocation of HMGB1 via LOC107986345/miR-128-3p/EPHB2 axis**. The previous results show that LSS does not only induce autophagy in endothelial cells, but also activates the LOC107986345/miR-128-3p/EPHB2 axis. Therefore, we speculated that LSS regulates endothelial cell autophagy through the LOC107986345/miR-128-3p/EPHB2 axis. First, we explored whether this ceRNA network regulates endothelial cell autophagy. Acidic lysosomes were stained with LTR fluorescent dye to detect autophagy in HAECs after different treatments. Compared with cells transfected with a control vector, pri-LOC107986345, sh-miR-128-3p, and pri-EPHB2 vectors significantly increased the red fluorescence intensity (Fig. 6a, b). Moreover, the effects of LOC107986345/ miR-128-3p/EPHB2 axis-induced autophagy were further investigated using transmission electron microscopy (TEM) images from HAECs. The results clearly show autophagosomes and autophagolysosomes in HAECs when treated with different vectors (Fig. S3t, u), and this mediation effect was due to activate LOC107986345/miR-128-3p/EPHB2 axis-induced autophagy rather than inhibition of the lysosomal degradation pathway (Fig. 6c–e). The above results suggested that LSS induces autophagy in endothelial cells through the LOC107986345/miR-128-3p/EPHB2 axis. To further explore whether autophagy mediated by LSS through the LOC107986345/miR-128-3p/EPHB2 axis can inhibit endothelial cell inflammation, we first used the monocyte adhesion experiments to show that adhesion of THP-1 to endothelial cells was significantly enhanced under TNF-α stimulation. However, this adhesion was significantly weakened by activation of the LOC107986345/miR-128-3p/EPHB2 axis (Fig.6f, g). We also measured the expression of related inflammatory factors and autophagy marker proteins. The results showed that LOC107986345/miR-128-3p/EPHB2 axis-mediated autophagy can inhibit endothelial cell inflammation (Fig. 6h, i). Moreover, the anti-inflammatory effect through activating the LOC107986345/miR-128-3p/EPHB2 axis was reduced when HAECs were transfected with HMGB1 siRNA, accompanied by a decrease in MAP1LC3B2 expression and an increase in SQSTM1/

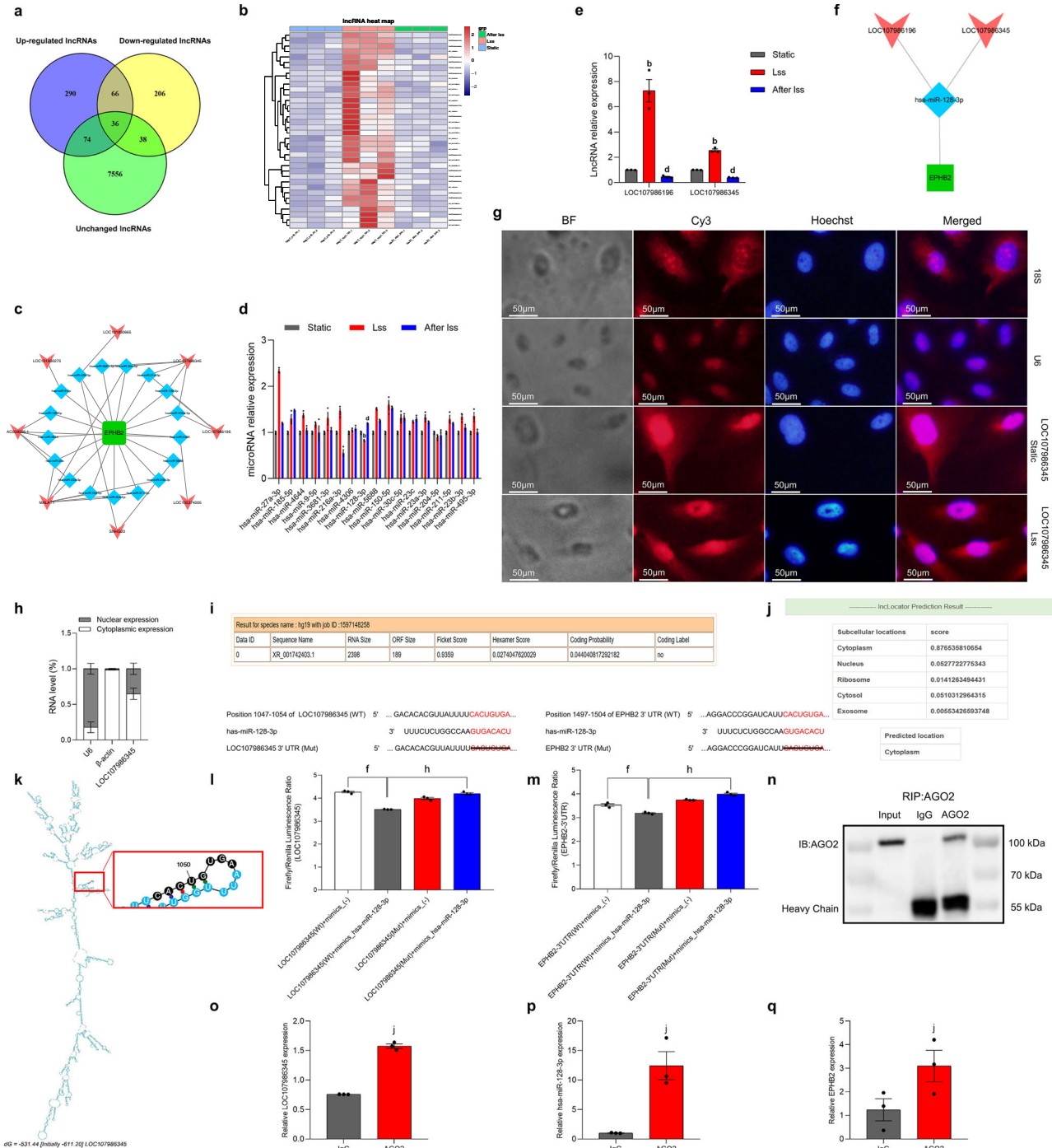

**Fig. 5 Construction of the LSS-related LOC107986345/miR-128-3p/EPHB2 network. a** DE lncRNA Venn diagram. **b** DE lncRNA heat map. **c** ceRNA network regulation diagram with EPHB2 as the core. **d** qRT-PCR shows the expression of 17 miRNAs. **e** qRT-PCR shows the expression of LOC107986345 and LOC107986196. **f** Construction of the LOC107986345/miR-128-3p/EPHB2 and LOC107986196/miR-128-3p/EPHB2 networks. **g** The FISH experiment shows the subcellular localization of LOC107986345 under a laser confocal microscope. 18S and U6 were used as internal controls. Cy3 label shows LOC107986345 labeled with lncRNA FISH probe and internal control (red), and Hoechst staining shows the cell nucleus (blue). **h** qRT-PCR shows the subcellular localization of LOC107986345. U6 was used as the nuclear gene reference, and β-actin was used as the cytoplasmic gene reference. **i** CPAT predicts the coding ability of LOC107986345. **j** LncLocator predicts the subcellular localization of LOC107986345. **k** Mfold web server predicts the secondary structure of LOC107986345. **l** Luciferase reporter gene experiment shows the targeting relationship between LOC107986345 and miR-128-3p. **m** Luciferase reporter gene experiment shows the targeting relationship between EPHB2 and miR-128-3p. **n**–**q**. Anti-AGO2 RIP assay shows the amounts of LOC107986345, miR-128-3p and EPHB2 in anti-AGO2 and anti-IgG immunoprecipitates. Data are presented as mean ± SEM of three independent experiments.

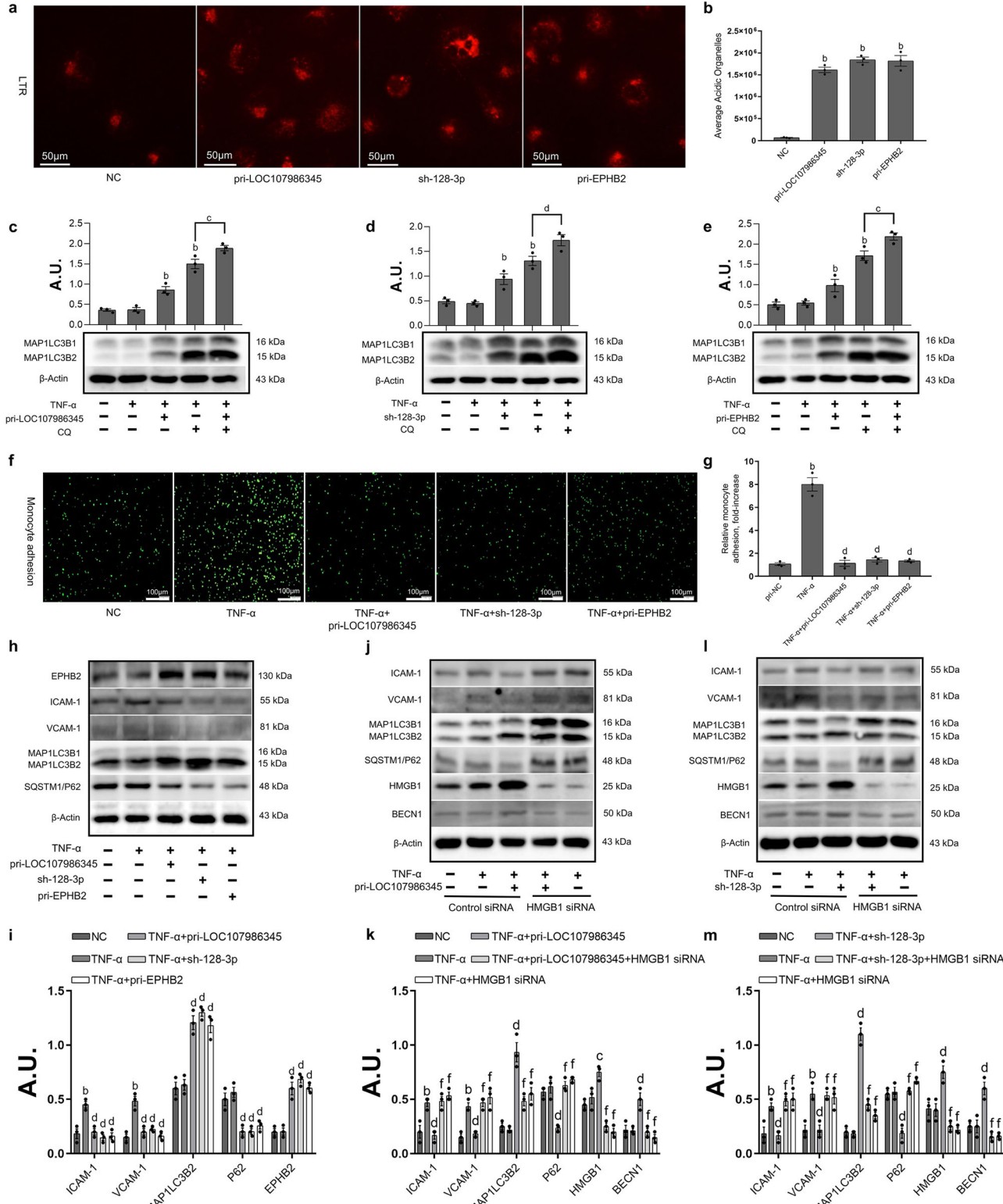

**Fig. 6 The inhibition of endothelial inflammatory by LSS depends on autophagy activation resulting from the nuclear translocation of HMGB1 via LOC107986345/miR-128-3p/EPHB2 axis. a**, **b** Confocal microscopy shows the effect of pri-LOC107986345, sh-miR-128-3p and pri-EPHB2 activated autophagy (HAECs stained with Lyso Tracker Red). **c**–**e** Western Blot shows the expression of MAP1LC3B2. **f**, **g** Confocal microscopy shows the effect of pri-LOC107986345, sh-miR-128-3p and pri-EPHB2 inhibited monocyte adhesion. **h**, **i** Western Blot shows the anti-inflammatory effect of pri-LOC107986345, sh-miR-128-3p and pri-EPHB2. **j**, **k** Western Blot shows the effect of HMGB1 siRNA on anti-inflammatory effect of pri-LOC107986345. **l**, **m** Western Blot shows the effect of HMGB1 siRNA on anti-inflammatory effect of sh-miR-128-3p. Data are presented as mean ± SEM of three independent experiments. $^{a}P < 0.05$, $^{b}P < 0.01$ vs. NC group; $^{c}P < 0.05$, $^{d}P < 0.01$ vs. TNF-α or TNF-α + CQ group. $^{e}P < 0.05$, $^{f}P < 0.01$ vs. TNF-α + pri-LOC107986345 or sh-miR-128-3p group. AU, arbitrary units.

P62 expression (Fig. 6j–m). This indicates that the inhibition of endothelial cell inflammatory by LSS depends on autophagy activation resulting from the nuclear translocation of HMGB1 via LOC107986345/miR-128-3p/EPHB2 axis.

## Discussion

The biggest difference between vascular endothelial cells and other endothelial cells is that they are subjected to LSS generated by blood flow. A large amount of evidence shows that LSS regulates endothelial cell homeostasis by affecting the expression of genes in endothelial cells[32,33]. Therefore, using LSS-sensitive genes as an entry point to study cardiovascular diseases may be a promising strategy to combat the occurrence and development of AS. AS is caused by long-term excessive inflammation in the blood vessel wall. Long-term exposure of vascular endothelial cells to inflammatory factors can cause the permeability of endothelial cells to increase, which leads to an increase in the probability of lipid entry and aggravates AS development[34,35]. It has been proven that autophagy is involved in inflammation regulation, starvation, cell development, senescence, oxidative stress, and programmed cell death, as well as pathophysiological processes, such as the immune response[6,7], especially in the context of vascular endothelial cell inflammation[36,37]. Studies have reported that LSS could inhibit endothelial cell inflammation by activating autophagy[12,14]. On this basis, we further proved that when the autophagy pathway was inhibited by 3-MA, the anti-inflammatory effect of LSS disappeared, indicating that LSS could inhibit the inflammatory response in an autophagy-dependent pathway in HAECs and HUVECs.

In this study, we used whole-transcriptome sequencing technology to study the changes in gene expression in HAECs before and after LSS to identify LSS-sensitive genes, and finally obtained a key gene-EPHB2. In recent years, the regulatory relationship between EPHB2 and autophagy has been widely reported in certain disease research[38,39]. However, few studies have reported that EPHB2 regulates vascular endothelial cell autophagy and inhibits endothelial cell inflammation under LSS. Our results indicated that LSS can activate endothelial cell autophagy through EPHB2, when EPHB2 expression was inhibited, the autophagy activation ability of LSS was greatly reduced. A large number of studies have shown that HMGB1, as an active regulator of autophagy, interacts with BECN1 in cells to induce the formation of autophagosomes[19,20,40]. We proved that there was an interaction between EPHB2 and HMGB1 through co-immunoprecipitation assay, and subsequent immuno-fluorescence experiments proved that HMGB1 can exit the nucleus and co-localize with EPHB2 under the action of LSS to induce autophagy in HAECs. Further research showed that overexpression of EPHB2 can activate autophagy and exert the same anti-inflammatory effect as LSS by inducing the nuclear translocation of HMGB1.

At present, the molecular mechanisms of lncRNAs in the regulation of AS have become a new focus of attention[41]. However, under the action of LSS with anti-AS properties, the expression and function of lncRNAs sensitive to LSS in vascular endothelial cells are still unclear. An increasing amount of evidence shows that some lncRNAs act as ceRNAs in regulating the biological functions of miRNAs[42,43]. In this study, we identified a novel LSS-sensitive lncRNA, LOC107986345, around EPHB2. Overexpression of LOC107986345 in HAECs significantly activated autophagy and inhibited enhanced monocyte adhesion and increased the expression of inflammatory factors after TNF-α stimulation. Then we used biosynthesis and the dual-luciferase reporter assay to confirm that miR-128-3p is the target of LOC107986345 and EPHB2. miR-128-3p is related to a variety of cardiovascular diseases[44,45]. Interestingly, we found that knocking down miR-128-3p in HAECs also significantly activated autophagy and inhibited inflammation. Our results showed that LSS can induce nuclear translocation of HMGB1 by activating LOC107986345/miR-128-3p/EPHB2 axis, the autophagy pathway of HAECs is rapidly activated after the nuclear translocation of HMGB1. However, when HMGB1 was inhibited, the aforementioned autophagy levels were significantly reduced, correspondingly, the anti-inflammatory ability of LSS also decreases. This indicates that the inhibition of endothelial cell inflammatory by LSS depends on autophagy activation resulting from the nuclear translocation of HMGB1 via activating LOC107986345/miR-128-3p/EPHB2 axis (Fig. S4).

In this study, we used immortalized HAECs and HUVECs, which have not differentiated and still express endothelial cell markers. Although they may not be representative of the physiological state, compared to primary endothelial cells, immortalized HAECs and HUVECs responded normally to LSS. The pro-atherosclerotic properties of oscillatory shear stress (OSS) have been recognized by the majority of scientific researchers[46]. Our next research is to explore the relationship between LOC107986345/miR-128-3p/EPHB2 axis and OSS. For example, does this axis down regulated in these pro-atherogenic flow condition? Does upregulating the identified axis would prevent OSS-induced endothelial cell inflammation? Validating the important role of LOC107986345/miR-128-3p/EPHB2 axis in the occurrence and development of atherosclerosis from another aspect.

## Methods

**LSS model**. The entire fluid-loading system consisted of a parallel flow chamber (NatureThink Life & Scientific Co., Ltd., China), a peristaltic pump (Longer Precision Pump Co., Ltd., Great Britain), a silicone hose, and a culture bottle (Fig. S2). Complete medium was pumped out by the peristaltic pump and entered the parallel flow chamber through the hose. LSS was applied to endothelial cells and then returned to the culture flask through the hose to form a closed circulatory system. The parallel flow chamber consisted of an upper chamber and a lower chamber. The lower chamber had a rectangular groove in the middle to place slides containing cells. There is a rectangular groove in the middle of the lower chamber for placing slides pretreated with poly-L-lysine solution (Solarbio, China). After the cells on the slides were at 80–90% confluence, the culture fluid flowed over the surface, and LSS at 12 dyn/cm² was applied to the cells. The apparatus was maintained at 37 °C throughout the duration of the experiment. The model of the peristaltic pump was C901, and the viscosity of the cell culture solution was 0.06 g/cm/s. To obtain an LSS of 12 dyn/cm², the liquid flow rate was 7.36 ml/min (http://www.naturethink.com/jql/jql.html).

**Cell culture and grouping**. The HAEC and HUVEC cell line were obtained from the Department of Pathogenobiology, Basic Medicine College of Jilin University (Changchun, China), these two kinds of cells were authenticated by cell line authentication service in Shanghai Subgene Institute and Shanghai Chuanqiu Biotechnology Co., LTD., respectively, and the identification results showed that the cells were not contaminated by known cell lines and were single-source cell lines. The HAECs and HUVECs were cultured in a moist atmosphere containing 5% carbon dioxide ($CO_2$) at 37 °C in Dulbecco's modified Eagle's medium (Gibco, NY, USA) or Iscove's modified Dulbecco's medium (Gibco, NY, USA) respectively, which supplemented with 10% fetal bovine serum (FBS; BD Biosciences, San Jose, CA, USA) and 1% penicillin-streptomycin (Sigma, St. Louis, MO, USA). THP-1 cells were maintained in Roswell Park Memorial Institute 1640 (Gibco) supplemented with 10% FBS at 37 °C with 5% $CO_2$. Adherent cells were digested with trypsin-EDTA digestion solution (0.25%, MRC, USA) and then passaged. HAECs used for whole-transcriptome sequencing were divided into three groups: the static-state group (Static group); the 12-dyn/cm² LSS group that continued to act on endothelial cells for 12 h (LSS group); and the 12-dyn/cm² LSS group, which after 12 h of treatment, were returned to steady-state culture for 24 h (After LSS group). TNF-α was present throughout the period of flow exposure in the experiment of LSS inhibiting the inflammatory response of endothelial cells stimulated by TNF-α (10 ng/ml).

**RNA isolation and library preparation**. Total RNA of the three groups of HAECs was extracted using TRIzol reagent according to the manufacturer's protocol. RNA purity and quantification were evaluated using the NanoDrop 2000 spectrophotometer (Thermo Fisher Scientific, Waltham, MA, USA). RNA integrity was

assessed using the Agilent 2100 Bioanalyzer (Agilent Technologies, Santa Clara, CA, USA). Then, the libraries were constructed using TruSeq Stranded Total RNA with Ribo-Zero Gold (Illumina, Cat. No. RS-122-2301) according to the manufacturer's instructions. RNA (1 μg) from each sample was used for library construction. First, ribosomal RNA (rRNA) was removed using Ribo-Zero Gold rRNA Removal Kit (Illumina), and RNA was fragmented using a fragment reagent. Subsequently, sequencing libraries were constructed using rRNA-depleted RNA. Finally, products were purified (Agencourt AMPure XP, BECKMAN COULTER), and library quality was assessed using the Agilent Bioanalyzer 2100 system.

**Whole-transcriptome RNA sequencing and DE RNA analysis**. The libraries were sequenced on an Illumina HiSeq X Ten platform, and 150-bp paired-end reads were generated. Approximately 98.19 M raw reads for each sample were generated. Raw data (raw reads) in fastq format were first processed using Trimmomatic software. In this step, clean data (clean reads) were obtained by removing reads containing adapter and ploy-N or low-quality reads from raw data. Then, approximately 13.47 M clean reads for each sample were retained for subsequent analyses. Sequencing reads were mapped to the human genome (GRCh38) using HISAT2. For mRNAs, the FPKM of each gene was calculated using Cufflinks, and the read counts of each gene were obtained using HTSeq-Count. For lncRNAs, the transcriptome from each dataset was assembled independently using the Cufflinks 2.0 program. All transcriptomes were pooled and merged to generate a final transcriptome using Cuffmerge. RNA expression profile data were analyzed with the DESeq2 package of R. By setting the $|\log2\text{FoldChange}| \geq 1$ and $P$ value $\leq 0.05$ as criteria, DE lncRNAs, and DE mRNAs were screened for subsequent analysis.

**PCA**. A PCA of mRNA expression profiles obtained by whole-transcriptome sequencing of the three groups of HAECs was performed using OmicShare Tools, which is a free online platform for data analysis (http://www.omicshare.com/tools).

**Short Time-series Expression Miner**. mRNA expression profiles obtained by whole-transcriptome sequencing of the three groups of HAECs were organized into different clusters based on expression patterns using Short Time-series Expression Miner[47].

**GO and KEGG enrichment analysis**. The Database for Annotation, Visualization, and Integrated Discovery (https://david.ncifcrf.gov/) and OmicShare Tools were used for the functional analysis of 96 DE mRNAs. The biological processes in the GO and KEGG pathways (adjusted $P$ values of < 0.05) were selected to analyze their biological function. The significant enrichment results were visualized using the ggplot2 package of R.

**Cell transfection**. All plasmids were purchased from the Public Protein/Plasmid Library (Nanjing, China), including LOC107986345 overexpression plasmid pLVX-Puro-LOC107986345 (pri-LOC107986345); miR-128-3p overexpression plasmid pLVX-Puro-miR-128 (pri-128-3p); EPHB2 overexpression plasmid pLenti-CMV-GFP-Puro-EPHB2 (pri-EPHB2); LOC107986345 knockdown plasmid pLKO.1-Puro-LOC107986345 (sh-LOC107986345-1, -2, and -3); miR-128-3p sponge plasmid pLKO.1-Puro-hsa-miR-128-3p (sh-128-3p); EPHB2 knockdown plasmid pPLK/GFP-Puro-EPHB2 (sh-EPHB2-1, -2, and -3); HMGB1 small interfering RNA (HMGB1 siRNA). pLVX-Puro and pLKO.1-Puro are overexpression and knockdown vectors, respectively; pLenti-CMV-GFP-Puro and pPLK/GFP-Puro are overexpression and knockdown vectors that express green fluorescence, respectively. The corresponding DNA fragment were generated by high fidelity PCR and cloned into the corresponding vector. The sequences used are shown in Table S3. Cells were transfected using X-tremeGENE HP DNA Transfection Reagent (Roche, Basel, Switzerland) according to the manufacturer's protocol. Twenty-four hours after cell transfection, the cells were analyzed for subsequent assays.

**qRT-PCR**. Total RNA (mRNA and lncRNA) was extracted from cultured cells using AxyPrep Total RNA Mini Preparation Kit (Corning, China), and miRNA was extracted using the miRNeasy Mini Kit (Qiagen, Germany). mRNA and lncRNA were reverse transcribed into complementary DNA (cDNA) using the HiScript II 1st Strand cDNA Synthesis Kit (Vazyme, China), and miRNA was reverse transcribed using the miRcute miRNA First-strand cDNA Synthesis kit (Tiangen, China). qRT-PCR to measure mRNA and lncRNA was performed using the ABI 7300 Plus Real-Time PCR System (Applied Biosystems, USA) using the FastStart Universal SYBR® Green Master Mix (Roche), and the miRcute Plus miRNA qPCR kit (Tiangen) was utilized to detect miRNA. The primers used for qRT-PCR assays are listed in Table S4. β-actin and U6 were used as endogenous controls for mRNA, lncRNA, and miRNA. Relative fold changes were calculated using the $2^{-\Delta\Delta Ct}$ method. All qRT-PCR assays were repeated three times.

**Immunoblotting**. Cells were homogenized in RIP assay (RIPA) buffer (Sigma-Aldrich, USA) with 1x Halt™ Protease and Phosphatase Inhibitor Single-Use Cocktail, EDTA-Free (Thermo Fisher Scientific). Cell lysate was centrifuged at 17,500 rcf for 15 min at 4 °C. The protein concentration was determined using the bicinchoninic kit (Sigma-Aldrich). Proteins were separated on sodium dodecyl sulfate-polyacrylamide gel electrophoresis gels, transferred to polyvinylidene fluoride membranes, and incubated with antibodies against the following targets: EPHB2, ICAM-1, VCAM-1, COX-2, MMP-9, MAP1LC3B2, SQSTM1/p62, AGO2, HMGB1, BECN1, Lamin B, β-actin, and GAPDH (Abcam, Great Britain). Horseradish-peroxidase AffiniPure goat anti-rabbit immunoglobulin G (IgG) was used as the secondary antibody. After adding an appropriate amount of electro-chemiluminescence chromogenic substrate, a chemiluminescence imaging system (GeneGnome XRQ, Great Britain) was used to detect western blot chemiluminescence, and the intensity was measured using Image J software.

**Co-immunoprecipitation**. Cells were homogenized in 0.5%Triton X-100(Beyotime Biotechnology, China) with 1x Halt™ Protease and Phosphatase Inhibitor Single-Use Cocktail, EDTA-Free (Thermo Fisher Scientific). Cell lysate was centrifuged at 12000 rpm for 20 min at 4 °C to extract clear lysates. For immunoprecipitation, incubate antibody (5 μg IgG or HMGB1)-magnetic beads protein A/G (Millipore, Germany) complex at 4 °C for 4 h with shaking. Add 50ug clear lysates to the precipitates after centrifugation. Incubate the cell lysates-antibody-magnetic beads complex with shaking at 4 °C overnight, then the beads were washed three times with lysis buffer and the precipitates were eluted with 100 μl 1 × SDS loading buffer at 100 °C for 10 min. Elutes and whole cell lysate (input) were resolved on SDS-PAGE followed by immunoblotting with antibodies.

**Immunofluorescence**. Fix the HAECs growing on the slides with 4% paraformaldehyde (Beyotime, China) for 30 min. Then the cells were permeabilized with 0.5% Triton X-100 for 15 min and blocked with 5% goat serum (Boster Biotechnology, China), followed by incubation with antibodies for overnight at 4 °C. Antibodies used here are rabbit anti-EPHB2 (Cell Signaling Technology, USA), mouse anti HMGB1 (Proteintech, USA), Alexa Fluor 488-labeled Goat Anti-Rabbit IgG(H + L) (Beyotime, China) and Cy3-labeled Goat Anti-Mouse IgG (H + L) (Beyotime, China). The Nuclear DNA was stained with Hoechst 33342 (ThermoFisher, USA). Images were acquired with a fluorescence microscope (Olympus IX71, Japan).

**LTR assay**. Lysosomal activity in HAECs was analyzed by LTR (Meilunbio, China) according to the manufacturer's instructions. 50 nM LTR containing working solution (preheated to 37 °C) was added to HAECs and incubated for 2 h under culture. Then, the staining solution was replaced with fresh medium, and the Radiance 2000 laser scanning confocal microscope (Nikon, Japan) was used for imaging.

**TEM**. Autophagosomes and autophagolysosomes in HAECs were observed by TEM. Cells were transfected with the corresponding plasmid in advance and after 24 h. Cells were trypsinized, which was stopped using fetal calf serum, and washed with phosphate-buffered saline. Cells were then collected by centrifugation, and 4% paraformaldehyde was added and allowed to stand at 4 °C for 12 h. Ultrathin sections were cut using an ultramicrotome, counterstained with 0.3% citrate, and examined using TEM (Hitachi, Japan).

**Nucleoplasm separation assay**. Cytoplasmic and nuclear RNAs in HAECs were separated, extracted, and purified using the Cytoplasmic & Nuclear RNA Purification kit (NORGEN, USA) in accordance with the manufacturer's protocol. The expression of LOC107986345, β-actin, and U6 was measured by qRT-PCR; Cytoplasmic and nuclear protein in HAECs were separated, extracted, and purified using the Nuclear/plasma protein extraction reagent kit (Solarbio, China) in accordance with the manufacturer's protocol.

**FISH**. The RNA FISH probe specifically designed for LOC107986345 was available from RiboBio Co., Ltd., and the experiment was performed according to the instructions for the Ribo™ FISH kit (RiboBio, Guangzhou, China). The human U6 FISH probe (Lnc 110101; RiboBio) and the human 18 S FISH probe (Lnc110102; RiboBio) were used as nuclear and cytoplasmic controls, respectively. All images were visualized under an inverted fluorescence microscope (Olympus IX71, Japan).

**Luciferase reporter assay**. The cDNA fragment containing the wild type or mutant LOC107986345 fragment and the $3'$ untranslated region of EPHB2 was subcloned downstream of the luciferase gene within the pGL3-control luciferase reporter vector (Promega, USA). The nucleotide sequences of all constructs were confirmed by DNA sequencing. Luciferase reporter plasmids plus mimics_miR-128-3p or mimics(-) were co-transfected into human embryonic kidney 293 cells using Lipofectamine 2000 (Invitrogen, Carlsbad, CA, USA). After 48 h of transfection, the dual-luciferase reporter assay kit (Promega) was utilized to measure luciferase activity.

**RNA binding protein immunoprecipitation (RIP)**. The RIP was performed using the Magna RIP™ RNA-Binding Protein Immunoprecipitation Kit (Millipore, USA) following the manufacturer's instructions. Briefly, $2 \times 10^7$ HAECs at 80–90%

confluence were harvested and lysed in RIP lysis buffer. After, 100 μL of cell lysate was incubated with RIP buffer containing magnetic beads conjugated with Rabbit Argonaute2 (AGO2) antibody (Abcam, Great Britain) or Rabbit IgG antibody (Abcam, Great Britain). Take 50 ul magnetic beads suspension and use Western blotting to detect the immunoprecipitation efficiency of AGO2. The specimens were incubated with Proteinase K (Solarbio, Beijing, China) to segregate immunoprecipitated RNAs. The extracted RNAs were further evaluated using qRT-PCR.

**Monocyte adhesion assay**. To assess the binding of THP-1 cells to HAECs, THP-1 cells were labeled with CFSE (5-and 6-Carboxyfluorescein diacetate, succinimidyl ester, USA), at a final concentration of 5 μM and then incubated with HAECs for another 4 h in a moist atmosphere containing 5% $CO_2$ at 37 ℃ (the adhesion of monocyte was done after HAECs exposure to LSS). Unbound cells in the dishes were removed by washing 5× with serum-free medium. Adherent cells were visualized under an inverted fluorescence microscope (Olympus IX71, Japan).

**Statistics and reproducibility**. All data in this study are presented as mean ± standard error of the mean. Statistical analyses were performed using IBM SPSS Statistics 20.0 software. The data in different groups were analyzed by one-way analysis of variance (ANOVA) followed by Tukey's post-hoc test. A $P$ value of <0.05 was considered statistically significant.

**Reporting summary**. Further information on research design is available in the Nature Research Reporting Summary linked to this article.

## Data availability

Sequencing data have been deposited in the GEO database, accession number GSE19970. Source data behind the graphs and charts in the paper can be found in Supplementary Data 1. Uncropped blots are provided as Supplementary Fig. S5. All other data can be obtained from the authors on reasonable request.

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

## Acknowledgements

This study received funding from the National Natural Science Foundation of China (Grant No. 82171553), Chinese Ministry of Foreign Affairs and Chinese Ministry of Education (Grant No. 105213000000200012) and the Fundamental Research Funds for the Central Universities, JLU.

## Author contributions

Q.M.: Investigation, Methodology, Project administration, and Writing-Original Draft; L.P.: Methodology, Formal analysis, Investigation and Project administration; M.Q.: Data Curation and Investigation; S.L.: Methodology and Project administration; B.S.: Software and Data Curation; Y.W.: Investigation and Software; B.L.: Resources, Validation, Writing-Review & Editing, and Supervision; F.L.: Conceptualization, Writing—Review & Editing, Supervision, and Funding acquisition.

## Competing interests

The authors declare no competing interests.
