## [Peer Review File · Communications Biology]

Reviewers' comments:

Reviewer #1 (Remarks to the Author):

The authors identified a novel Laminar shear stress (LSS)-sensitive lncRNA, LOC10798635, and demonstrated that the inhibition of endothelial inflammatory by LSS, it depends on autophagy activation. Autophagy is activated by LOC107986345/miR-128-3p/EPHB2 axis which induces the nuclear translocation of high mobility group box-1 (HMGB1), which interacts with Beclin-1 and finally leads to autophagy in endothelial cells.

The lncRNA LOC10798634 and the mechanisms describe in the study are novel; however, there are technical and conceptual issues that make the study and the experiments of difficult interpretation.

Major issues:

Cell transfection: the primary endothelial cells such as HUVECs and HAEC are cells very difficult to transfect and the maximum efficiency using lipid-based transfection reagents is around 20%. All the experiments in this study are performed using this transfection method with high-molecular-weight plasmids without performing any selections. Based on this, the identified phenotype is present only in small population of the cells analysed. Moreover, there aren't information how these plasmids are cloned and how they have been tested. For example, the sponge vector for miR128b has a minimal effect on miR128 expression with 10% or less downregulation (figure 4J). The authors did not provide any info on LOC10798634. Location and more info are missing. The authors did not provide any in vitro transcription data or RNA structure data.

Figure 2: it is very difficult to understand this picture. The figure in a manuscript must be self-explanatory. There are no indication of which mRNAs and lncRNAs are represented.

Figure 4: There is no correlation between the overexpression of LOC10798634 and upregulation of EPHB2 (figure 4 A-C). How the authors identified the binding site for miR128 in LOC10798634? There are no western blots for Ago2 showing the successful pull down (figure 4 O-Q). Moreover, the authors should pull down for biotinylated LOC10798634 and check if Ago2 and miR128 are interacting with LOC10798634.

Figure 5: in the panel J the localization of GFP-LC3 in control is very unusual. Moreover, the LC3 positive dots are very big compared to the cell nucleus. In Panel F and G the experiments do not highlight the consequentiality between LOC10798634, miR128b and EPHB2. The authors should overexpress LOC10798634 and knockdown EPHB2. Moreover, in the western blot in panel F the overexpression of EPHB2 don't increase EPHB2 protein level, probably for the low efficacy in the transfection.

Figure 6: Again, localization and puncta in RFP-LC3-GFP are not detectable in the picture in panel A.

Figure 7: How does EPHB2 regulate HMGB-1 translocation? Moreover, the images in panel D don't show a nuclear localization in static and cytoplasmic localization in LSS.

Uncut western blot must be provided for all the experiments.

Reviewer #2 (Remarks to the Author):

Reviewing "Laminar shear stress-sensitive long non-coding RNA, LOC10798635, inhibits inflammation in an autophagy-dependent pathway in human aortic endothelial cells via regulation of EPHB2"

Meng and colleague identified a new lncRNA, LOC10798635, regulated by laminar shear stress with anti-inflammatory properties by regulating autophagy in ECs.

The work presented is well done with some improvement that could be made (see further comments). Despite the identification of this new "LOC", the manuscript lack of novelty as the link between LSS and autophagy, the link between autophagy and atherosclerosis, the role of ephB2 in autophagy pathways have been already shown. Their novelty is that they identify that EPHB2 is related to autophagy in endothelial cell which have been shown only in epithelial cells and cancer before.

The introduction is missing some important references in the field and the discussion should be

reworked to better highlight what is specifically new in this work. Additionally the perspective on treatment should be emphasize if possible because this would raise the interest of the discovery.

Introduction

Meng and colleagues tried to introduce all the components of their story in their introduction. Despite this effort, the introduction remains unclear and without fluidity. Especially the part from line 85 until 96 which sounds more like a list of information rather than a constructed thought. Especially the introduction of miR128-3p is not well linked with the previous part of the intro, why talking about this miR specifically ?

Why talking about myofibroblast (ref16-17) in the context of AS ?

The literature on autophagy and atherosclerosis is subsequent but some important publication in the area are missing in this introduction which would help the reader to have a better overview and a better understanding for the importance of the work. Additionally the link between laminar flow and autophagy is known, as well as its atheroprotective effect. This should be clearly stated. Example of articles which should/could be cited :

- Yao et al. Laminar Shear Stress Promotes Vascular Endothelial Cell Autophagy Through Upregulation with Rab4. DOI: 10.1089/dna.2015.3041
- Liu et al. Shear stress regulates endothelial cell autophagy via redox regulation and Sirt1 expression. DOI: 10.1038/cddis.2015.193
- Vion et al. Autophagy is required for endothelial cell alignment and athero-protection under physiological blood flow. DOI: 10.1073/pnas.1702223114
- Santovito et al. Noncanonical inhibition of caspase-3 by a nuclear microRNA confers endothelial protection by autophagy in atherosclerosis. DOI: 10.1126/scitranslmed.aaz2294
- Yuan et al . Laminar flow inhibits the Hippo/YAP pathway via autophagy and SIRT1-mediated deacetylation against atherosclerosis. DOI: 10.1038/s41419-020-2343-1

Also the link between lncRNA, autophagy and atherosclerosis could be more fluent by using adequate bibliography and citing recent reviews on the subject :

- Ren et al. LncRNA-modulated autophagy in plaque cells: a new paradigm of gene regulation in atherosclerosis? DOI: 10.18632/aging.103786
- Fang et al . Recent advances on the roles of LncRNAs in cardiovascular disease. DOI: 10.1111/jcmm.15880

Ref 5 does not seem to fit the corresponding test.

Ref 8 does not seem to fit the corresponding test.

Ref 23 does not seem to fit the corresponding test (oscillatory flow not LSS).

Results

Figure 1:

- E to H, what is the point of showing shear stress induced autophagy in cells which are not flow sensitive ?

If this has a clear role, it should be better explained in the text.

Flow sensitive epithelial cells (urinary tract for example) would have make more sense here.

- 3-MA is a non-specific inhibitor, which could have non-autophagy related effects. Can the authors confirm their results by using siRNA strategy to block autophagy pathway ?

Figure 2 :

- As it is now this figure is not very informative and difficult to read at first sight. I would include the figure S2 in it and detail more with title and legend for each part of the figure (the legend is not enough).

Figure 3 :

- Based on the text written lines 159 to 163 it is difficult to understand what gene the authors selected with which criteria. Why keeping the gene with no change upon LSS ?

The text in this figure it too small, almost not readable. For example, the GO analysis presentation is nice but difficult to read (too small and not good positioning of the text).

To me, figure H,I and J are not essential to show in the main figure as they do not show strico-sensus a result but rather a way of organizing already found data. This would live more space to the rest and allow the authors to increase figure and text size.

- Figure 3M : the fold change for the miR-128-3p, even if significant, seems very small, how confident the author is that this small decrease in miR-128-3p is responsible for the upregulation of EPHB2 under LSS ?

How the important changes in the corresponding lncRNA can translate in such a little effect on miR-128-3p expression ?

Figure 4 :

- Figure 4B-C. Can the authors comments on the fact that the lower dose of LOC107986345 lead to the best mir down regulation (and subsequent EPHB2 upregulation) ?

- Figure 4F : the decrease in EPHB2 expression (mRNA level) is small, even if significant, especially for RNA expression. Could the authors also confirm that this mRNA decrease lead to a decrease at the protein level too ?

Same comment for 4I and 4L, western blot of EPHB2 would strengthen the claim of the authors.

- The figure M to Q showing the direct interaction are nice. I just have one comment, in the text associate with figure 3 it sounds like the "axe" identified between loc/mir/Ephb2 is one way (from loc to ephb2) therefor when seeing figure 4H and 4K before the interaction one raises useless questions.

I would either make clear in the figure 3 that the "axe" has no sequential order or I would move the interaction figures before the qPCR 4H and 4K.

Figure 5 :

- Did the authors tried to do FISH on ECs exposed to flow ? This could affect the localization of LOC107986345.

- Line 240. "Hence, negative regulation of EPHB2 by sponging miR-128-3p was attenuated by cytoplasmic LOC107986345 acting as a ceRNA". I don't understand how the author jump to that conclusion just by looking where is LOC (especially as the FISH showed it was also nuclear)

- How does the authors explain the massive increase in LC3 I in the pri-LOC and pri-miR conditions ?

- Figure 5 would have more power if the authors could also perform some of the experiments (if not all) with ECs exposed to flow and not under static condition as the claim of the authors is that this axis is the one regulated autophagy under flow.

Figure 6 :

- In the same line as for the figure 5. The authors showed here that their identified axis, when activated, prevent TNFa induced inflammation but the link with flow is not direct. Some of the experiment such as the monocyte adhesion should be performed under flow.

Figure 7 :

- Figure 7D : The image 3 (after LSS) seems out of focus. Please correct. Also, the nuclear translocation is difficult to assess in those images and should be properly quantified (ratio between the cytoplasmic and the nuclear signal).

- the legend of the figure is not well written. It should not state results but just describe the graph/images. "L-Q. Observe the effect of activating LOC107986345/miR-128-3p/EPHB2 axis to inhibit inflammation when HAECs were transfected with HMGB1 siRNA" is not a correct sentence (it appears several times, all the legend should be checked carefully).

- graphs O to Q are really difficult to follow. The authors should keep what is important for their message in the main and put the rest in supplemental to have lighter graphs.

- how does the authors explain the massive increase in LC3 I in the siHMGB1 condition ? (same for GA treatment)

Mat and Meth

I didn't carefully read this part as there was already plenty of comments that need answers. It sounds wired to have them before the discussion.

Discussion

The fact that LSS prevent inflammation of ECS through autophagy is already known and should be state by the authors on top of their observation.

There are also some issues with the English (verbs conjugation mostly) in the discussion.

Overall, they should better state what is not new and what is new.

General questions

Low or oscillatory flow also drive inflammation of the ECs. It is one of the main trigger of AS formation, even before or without any associated inflammatory molecules effect. Does this axis (LOC/miR/EPHB2) down regulated in these pro-atherogenic flow condition ? Does upregulating the identified axis would prevent oscillatory-flow-induced inflammation ?

Reviewer #3 (Remarks to the Author):

The authors have investigated the mechanisms by which LSS (12dynes/cm for 12h) promotes anti-inflammatory signalling in cultured human endothelial cells. They demonstrate that LSS reduces TNFa-induced inflammation via increased autophagy. They also deduced from RNAseq data that EPHB2 can be induced by LSS and that this may regulate autophagic flux. Lnc and miRNA that could influence EPHB2 expression were identified and validated in EC exposed to LSS. Evidence is presented suggesting that LOC10798635 acts as a competitive endogenous RNA, reducing the action of miRNA-128-3p and so increasing the expression and activity of EPHB2. This leads to increased HMGB nuclear localisation and regulation of key autophagic components.

This is a very comprehensive study with a lot of data presented and overall I think this is a strong manuscript and contributes to our knowledge of how LSS regulates cytoprotective responses. There are a few issues with the details in the methods/figure legends as outlined below:

The authors should provide more detail on how the cells were obtained and what ethical guidelines were followed. They should also provide information on what passage the cells were used at and what matrix (if any) cells were seeded onto. Please state whether cells were confluent at the onset of flow exposure.

Relating to cells and independent experiments – when it states ‘three independent experiments’ does this mean that cells were used from three different donors??

The authors should explain the rationale for returning EC to static conditions for 24h after exposure to LSS; it is not clear how this relates to the physiological setting where flow is continuous.

Figure 1. Please clarify the MAP1LC3B1 blot in the figure legend – was this used to quantify levels of LC3B2??

It is not clear from the figure legends or main text how the TNF-treated experimental group was carried out – was TNF added for 12h and then removed and the cultures maintained under static conditions? Was TNFa present throughout the period of flow exposure? The data showing that LSS reduces TNFa-induced inflammation via activation of autophagy is interesting however the physiological relevance is unclear as it seems as though TNFa was applied to static cultures and then LSS was applied, it would be more informative to conduct this experiment where TNFa is added to cells in the presence of flow and assess the impact of autophagy inhibition.

Comment on why only lower doses of Lnc increase EPHB2 expression (Fig 4C) whilst all doses affect 128-3p expression.

The authors should comment on the limitation that only LSS was studied relative to static culture - this is important considering that atherosclerotic lesions develop typically in regions of the vasculature exposed to non-uniform/disturbed flow.

The authors should also comment on the duration of flow exposure and why the 12h timepoint, when EC are still adapting to shear stress, was chosen. It would be beneficial to show the involvement of this pathway under chronic conditions (over 48h)

The concluding statement is a bit of a leap with regards to the potential to 'prevent and treat' atherosclerosis considering that this is solely based on in vitro work. The authors should comment on this limitation and comment on future work required to support this statement.

Minor comments

Shear stress should be defined as the frictional force per unit area

Many of the individual figure panels are too small

Figure 2 should be included as a supplementary figure

Reviewers' comments:

Reviewer #1 (Remarks to the Author):

The authors identified a novel Laminar shear stress (LSS)-sensitive lncRNA, LOC10798635, and demonstrated that the inhibition of endothelial inflammatory by LSS, it depends on autophagy activation. Autophagy is activated by LOC107986345/miR-128-3p/EPHB2 axis which induces the nuclear translocation of high mobility group box-1 (HMGB1), which interacts with Beclin-1 and finally leads to autophagy in endothelial cells.

The lncRNA LOC10798634 and the mechanisms describe in the study are novel; however, there are technical and conceptual issues that make the study and the experiments of difficult interpretation.

Major issues:

Cell transfection: the primary endothelial cells such as HUVECs and HAEC are cells very difficult to transfect and the maximum efficiency using lipid-based transfection reagents is around 20%. All the experiments in this study are performed using this transfection method with high-molecular-weight plasmids without performing any selections. Based on this, the identified phenotype is present only in small population of the cells analysed. Moreover, there aren't information how these plasmids are cloned and how they have been tested. For example, the sponge vector for miR128b has a minimal effect on miR128 expression with 10% or less downregulation (figure 4J).

Thank you very much for your professional comments. Indeed, the transfection efficiency of primary endothelial cells is very low. However, the cells used in this study are cell lines. The HAEC and HUVEC cell line were obtained from the Department of Pathogenobiology, Basic Medicine College of Jilin University (Changchun, China), these two kinds of cells were authenticated by cell line authentication service in Shanghai Subgene Institute and Shanghai Chuanqiu Biotechnology Co., LTD., respectively, and the identification results that the cells were not contaminated by known cell lines and were single source cell lines. All plasmids were purchased from the Public Protein/Plasmid Library (Nanjing, China). The plasmid cloning method is as follows: pLVX-Puro and pLKO.1-Puro are overexpression and knockdown vectors, respectively; pLenti-CMV-GFP-Puro and pPLK/GFP-Puro are overexpression and knockdown vectors that express green fluorescence, respectively. The corresponding DNA fragment were generated by high fidelity PCR and cloned into the corresponding vector, and this part was added to the Materials and Methods and highlighted in yellow. When we used different concentrations of the sponge vector for miR128-3p to knock down its expression, we found that the knock-down concentration of 0.02 μ g/ μ l was meaningful (pvalue=0.022), and successfully used this concentration to verify the expression of LOC107986345 and EPHB2, so we did not pursue the knockdown efficiency further.

The authors did not provide any info on LOC10798634. Location and more info are missing. The authors did not provide any in vitro transcription data or RNA structure data.

In the revised manuscript, we provide the location and sequence information of LOC107986345 (Table 6). Then we used nuclear and cytoplasmic RNA isolation technology to extract nuclear and cytoplasmic RNA separately and detect the expression of LOC107986345 by qRT-PCR. The results showed that LOC107986345 was mainly distributed in the cytoplasm (Fig. 5G). Coding potential assessment tool (CPAT) was used to predict the coding ability of LOC107986345. The prediction results showed that LOC107986345 does not have a protein-coding ability (Fig. 5H). The distribution of LOC107986345 was predicted by the lncRNA subcellular location prediction website, IncLocator. The results showed that the score in the cytoplasm was 0.88 (Fig. 5I). Fluorescence in-situ hybridization (FISH) was executed to identify the localization of LOC107986345 in HAECs, the results indicate that LOC107986345 was located in both the cytoplasm and nucleus (Fig. 5J). Finally, we use the Mfold web server (<http://www.unafold.org/mfold/applications/rna-folding-form.php>) to predict the secondary structure of LOC107986345 and use the dual-luciferase reporter assays to verify that there was a binding site between LOC107986345 and miR-128-3p (Figs. 5K-L).

Figure 2: it is very difficult to understand this picture. The figure in a manuscript must be self-explanatory. There are no indication of which mRNAs and lncRNAs are represented.

In the revised manuscript, we deleted the trend analysis of lncRNAs and displayed a more intuitive trend analysis of mRNAs in Fig. 2B.

Figure 4: There is no correlation between the overexpression of LOC10798634 and upregulation of EPHB2 (figure 4 A-C). How the authors identified the binding site for miR128 in LOC10798634? There are no western blots for Ago2 showing the successful pull down (figure 4 O-Q). Moreover, the authors should pull down for biotinylated LOC10798634 and check if Ago2 and miR128 are interacting with LOC10798634.

We have made major adjustments to the manuscript structure to better explain the interaction between LOC107986196, miR-128-3p, and EPHB2. We use the online database (miRDB) to identify the binding site between miR128 and LOC10798634, and the results show that the position 1047-1054 of LOC107986345 has a binding site with has-miR-128-3p, the results are as follows:

MicroRNA and Target Gene Description:

miRNA Name	hsa-miR-128-3p	miRNA Sequence	UCACAGUGAACCGUCUCUUU
Previous Name	hsa-miR-128a		
Target Score	79	Seed Location	1047
Target Length	2398		

Custom Target Sequence

```
1 aaatttggc ttagcaatt ttcaatttt ttacgatac ttatatoca atggagcaac
61 caaatatgaa tccfagaat gaagatgaa aatgaatatt ttaatcaat atcgagtgt
121 tctttcaaca ttttagttt tagttttac tgaatatata aagttacata attgtgng
181 gaataatatt tgcagagaaa tttttaaaaa ttacaaaag gaaaaagaaa ctttaccat
241 gagctatana cattgaanaa ctgtgttaa gtgattttaa taaagccaaa ccaacactt
301 cagcagaaac taatggcag catctgatt caagctgata catagtatt agtgggtgc
361 tcaactttac cctcaattt acctgggac ttgcatttt cttctattt ttacgtttt
421 tttttttt tcttttga agagtaatt tttaggaaa aaaaaaacat tttctcat
481 aggtcgattt aanaattgg ccttacctt atctctctc tcaactcna tcaactatg
541 aatgggtcaa cactctttc tgtgaagac cagatagtaa atattttagt cttgcagc
601 catatgctc ctgtgcccag ctttccatt gtgtgtgaa agcagccata gacaacag
661 aanaataaa gtgtaactg tcaataaaa acagatgata tgtgaattt agctcacaga
721 gtttagcttg ctgcccctc agagacctt tggatgaaa gcttctgag agaggaatt
781 tttgtttt ttgctaaagt tctagctca gtaactaaa tagtgcctg caagtagta
841 ttgatgata ttgaacctg tgaacatac acctaaaata aacatttgg caagatcac
901 tactacaaa ttgaaaac acttgctcc catagaaac aaagcttcc tgaataita
961 attatttgc ctgatgata attactgct ctgatgat agagtaatt tattttcaa
1021 ttactcagc gacacacgt tattttcaat gtgatttgg ttaaatgaa aagatttctt
1081 gctctaactc ctgataaac ctttatgaa taccatactc ttactcttt tttaatcca
```

In addition, we supplemented the western blots results of RIP and proved that Ago2 was successfully pulled down (Fig. 5N). However, the RNA-protein pull-down assay did not detect AGO2 in the protein complex binding with LOC10798634. We speculate that the content of AGO2 in the complex is too low to be detected by western blots.

Figure 5: in the panel J the localization of GFP-LC3 in control is very unusual. Moreover, the LC3 positive dots are very big compared to the cell nucleus. In Panel F and G the experiments do not highlight the consequentiality between LOC10798634, miR128b and EPHB2. The authors should overexpress LOC10798634 and knockdown EPHB2. Moreover, in the western blot in panel F the overexpression of EPHB2 don't increase EPHB2 protein level, probably for the low efficacy in the transfection.

In the revised manuscript, we have adjusted the structure of the manuscript and deleted this part of the results that you are worried about.

Figure 6: Again, localization and puncta in RFP-LC3-GFP are not detectable in the picture in panel A.

In the revised manuscript, we have adjusted the structure of the manuscript and deleted this part of the result that you are worried about. In addition, we use mRFP-GFP-LC3 adenovirus to monitor autophagy flux under the action of LSS, the results showed the localization and puncta are detectable (Fig. 3F).

Figure 7: How does EPHB2 regulate HMGB-1 translocation? Moreover, the images in panel D don't show a nuclear localization in static and cytoplasmic localization in LSS.

In order to explore the relationship between EPHB2 and HMGB1 in HAECs stimulated by LSS, co-IP assay was used to observe whether they can interact, and we were pleasantly surprised to find that there was an interaction between EPHB2 and HMGB1 (Fig. 3K). Subsequent immunofluorescence experiments proved that HMGB1 can exit the nucleus and co-localize with EPHB2 under the action of LSS (Fig. 3M), which indicated LSS could enhance the

EPHB2-mediated nuclear translocation of HMGB1. In addition, we did the GFP-HMGB1 experiment again and showed the percentage of cytoplasmic HMGB1 positive cells (Figs. 3J-K).

Uncut western blot must be provided for all the experiments.

We have provided all uncut western blot images in Supplementary Data.

Reviewer #2 (Remarks to the Author):

Reviewing “Laminar shear stress-sensitive long non-coding RNA, LOC10798635, inhibits inflammation in an autophagy-dependent pathway in human aortic endothelial cells via regulation of EPHB2”

Meng and colleague identified a new lncRNA, LOC10798635, regulated by laminar shear stress with anti-inflammatory properties by regulating autophagy in ECs.

The work presented is well done with some improvement that could be made (see further comments). Despite the identification of this new “LOC”, the manuscript lacks of novelty as the link between LSS and autophagy, the link between autophagy and atherosclerosis, the role of ephB2 in autophagy pathways have been already shown. Their novelty is that they identify that EPHB2 is related to autophagy in endothelial cell which have been shown only in epithelial cells and cancer before.

The introduction is missing some important references in the field and the discussion should be reworked to better highlight what is specifically new in this work. Additionally, the perspective on treatment should be emphasize if possible because this would raise the interest of the discovery.

Introduction

Meng and colleagues tried to introduce all the components of their story in their introduction. Despite this effort, the introduction remains unclear and without fluidity. Especially the part from line 85 until 96 which sounds more like a list of information rather than a constructed thought. Especially the introduction of miR128-3p is not well linked with the previous part of the intro, why talking about this miR specifically?

Why talking about myofibroblast (ref16-17) in the context of AS?

The literature on autophagy and atherosclerosis is subsequent but some important publication in

the area are missing in this introduction which would help the reader to have a better overview and a better understanding for the importance of the work. Additionally, the link between laminar flow and autophagy is known, as well as its atheroprotective effect. This should be clearly stated.

Example of articles which should/could be cited:

- Yao et al. Laminar Shear Stress Promotes Vascular Endothelial Cell Autophagy Through Upregulation with Rab4. DOI: 10.1089/dna.2015.3041
- Liu et al. Shear stress regulates endothelial cell autophagy via redox regulation and Sirt1 expression. DOI: 10.1038/cddis.2015.193
- Vion et al. Autophagy is required for endothelial cell alignment and athero-protection under physiological blood flow. DOI: 10.1073/pnas.1702223114
- Santovito et al. Noncanonical inhibition of caspase-3 by a nuclear microRNA confers endothelial protection by autophagy in atherosclerosis. DOI: 10.1126/scitranslmed.aaz2294
- Yuan et al. Laminar flow inhibits the Hippo/YAP pathway via autophagy and SIRT1-mediated deacetylation against atherosclerosis. DOI: 10.1038/s41419-020-2343-1

Also the link between lncRNA, autophagy and atherosclerosis could be more fluent by using adequate bibliography and citing recent reviews on the subject:

- Ren et al. LncRNA-modulated autophagy in plaque cells: a new paradigm of gene regulation in atherosclerosis? DOI: 10.18632/aging.103786
- Fang et al. Recent advances on the roles of LncRNAs in cardiovascular disease. DOI: 10.1111/jcmm.15880

Ref 5 does not seem to fit the corresponding test.

Ref 8 does not seem to fit the corresponding test.

Ref 23 does not seem to fit the corresponding test (oscillatory flow not LSS).

Thanks for your careful comments, we have made major revisions to the Introduction section of this manuscript according to your suggestions, and deleted and added the corresponding references.

Results

Figure 1:

- E to H, what is the point of showing shear stress induced autophagy in cells which are not flow sensitive?

If this has a clear role, it should be better explained in the text.

Flow sensitive epithelial cells (urinary tract for example) would have make more sense here.

- 3-MA is a non-specific inhibitor, which could have non-autophagy related effects. Can the authors confirm their results by using siRNA strategy to block autophagy pathway?

When initially designing this part of the experiment, we just wanted to prove that LSS can specifically induce autophagy in vascular endothelial cells such as HAECs and HUVECs, while other non-vascular endothelial cells do not. We did not consider other cells that may be sensitive to fluids, such as urothelial cells. Your suggestion is very good, we will refer to it in the follow-up

research. In response to your suggestion, we have supplemented corresponding experiment results in Figure S2 by transfection the siRNA of BECN1 into HAECs and HUVECs to better prove our conclusion.

Figure 2:

- As it is now this figure is not very informative and difficult to read at first sight. I would include the figure S2 in it and detail more with title and legend for each part of the figure (the legend is not enough).

In the revised manuscript, we deleted the trend analysis of lncRNAs and displayed a more intuitive trend analysis of mRNAs in Fig. 2B.

Figure 3:

- Based on the text written lines 159 to 163 it is difficult to understand what gene the authors selected with which criteria. Why keeping the gene with no change upon LSS?

The text in this figure is too small, almost not readable. For example, the GO analysis presentation is nice but difficult to read (too small and not good positioning of the text).

To me, figure H, I and J are not essential to show in the main figure as they do not show strico-sensus a result but rather a way of organizing already found data. This would live more space to the rest and allow the authors to increase figure and text size.

- Figure 3M: the fold change for the miR-128-3p, even if significant, seems very small, how confident the author is that this small decrease in miR-128-3p is responsible for the upregulation of EPHB2 under LSS?

How the important changes in the corresponding lncRNA can translate in such a little effect on miR-128-3p expression?

We made a mistake in the description of the selected cross gene and corrected it in the revised manuscript, and highlighted in yellow. The correct description is: We took the intersection of upregulated genes after LSS, downregulated genes after restoration of steady-state culture, and genes for which expression did not change in Static and After LSS group, finally got 96 differentially expressed (DE) mRNAs.

In the revised manuscript, we have adjusted the structure of the manuscript and increased the figure and text size as you mentioned.

The fold change for the miR-128-3p looks a bit small, however, the pvalue is already less than 0.001, which may not be obvious to the naked eye in the figure. Regarding the function of miR-128-3p, we carried out a lot of experiments to support our hypothesis. The results showed that miR-128-3p indeed played an important role in the LOC107986345/miR-128-3p/EPHB2 axis.

Figure 4:

- Figure 4B-C. Can the authors comments on the fact that the lower dose of LOC107986345 lead to the best mir down regulation (and subsequent EPHB2 upregulation)?

- Figure 4F: the decrease in EPHB2 expression (mRNA level) is small, even if significant, especially for RNA expression. Could the authors also confirm that this mRNA decrease lead to a decrease at the protein level too?

Same comment for 4I and 4L, western blot of EPHB2 would strengthen the claim of the authors.

- The figure M to Q showing the direct interaction are nice. I just have one comment, in the text associate with figure 3 it sounds like the “axe” identified between loc/mir/Ephb2 is one way (from loc to ephb2) therefore when seeing figure 4H and 4K before the interaction one raises useless questions.

I would either make clear in the figure 3 that the “axe” has no sequential order or I would move the interaction figures before the qPCR 4H and 4K.

We found this interesting phenomenon in our experiments. It is not that the higher the expression of LOC107986345, the lower the expression of miR-128-3p. We think that this mechanism of ceRNA in cells interacts with each other in a range of concentrations. When the concentration of LncRNA is very high, the interaction range is exceeded, so the expression of microRNA will not decrease as expected. In this study, when LOC107986345 reached an appropriate concentration (0.001 $\mu\text{g}/\mu\text{L}$), the expression of miR-128-3p decreased the most, and the expression of EPHB2 also increased the most.

We have added the corresponding western blot results of EPHB2 to support our conclusion in Fig. S2.

In response to your suggestion, we have reordered the results of this part in the revised manuscript.

Figure 5:

- Did the authors tried to do FISH on ECs exposed to flow? This could affect the localization of LOC107986345.

- Line 240. “Hence, negative regulation of EPHB2 by sponging miR-128-3p was attenuated by cytoplasmic LOC107986345 acting as a ceRNA”. I don’t understand how the author jump to that conclusion just by looking where is LOC (especially as the FISH showed it was also nuclear)

- How does the authors explain the massive increase in LC3 I in the pri-LOC and pri-miR conditions?

- Figure 5 would have more power if the authors could also perform some of the experiments (if not all) with ECs exposed to flow and not under static condition as the claim of the authors is that this axis is the one regulated autophagy under flow.

We added FISH on ECs exposed to LSS in Fig.5G in our revised manuscript, the results indicated that LOC107986345 was located in both the cytoplasm and nucleus whether the HAECs are in static or under the action of LSS.

We have made major adjustments to the manuscript, revised the questions you mentioned, and supplemented the corresponding experiments in Fig.3C, D and F.

Figure 6:

- In the same line as for the figure 5. The authors showed here that their identified axis, when activated, prevent TNFa induced inflammation but the link with flow is not direct. Some of the experiment such as the monocyte adhesion should be performed under flow.

Western blot and monocyte adhesion experiments have been supplemented in the revised manuscript to further prove the association between them in Fig.S2E-G.

Figure 7:

- Figure 7D: The image 3 (after LSS) seems out of focus. Please correct. Also, the nuclear translocation is difficult to assess in those images and should be properly quantified (ratio between the cytoplasmic and the nuclear signal).
- the legend of the figure is not well written. It should not state results but just describe the graph/images. "L-Q. Observe the effect of activating LOC107986345/miR-128-3p/EPHB2 axis to inhibit inflammation when HAECs were transfected with HMGB1 siRNA" is not a correct sentence (it appears several times, all the legend should be checked carefully).
- graphs O to Q are really difficult to follow. The authors should keep what is important for their message in the main and put the rest in supplemental to have lighter graphs.
- how does the authors explain the massive increase in LC3 I in the siHMGB1 condition? (same for GA treatment)

We have corrected this part of the result and performed a reasonable quantification in Fig.3J-K, and we also adjusted the legend of the figure.

We have made major adjustments to the structure of the manuscript, and have moved graphs O to Q to Fig.S2.

We speculate that when si-HMGB1 was used to inhibit its expression, the flow of autophagy in the cell will be hindered. There is no way to smoothly transform LC3 I into LC3 II, which leads to the accumulation of LC3 I, so a large amount of LC3 I will appear.

Mat and Meth

I didn't carefully read this part as there was already plenty of comments that need answers. It sounds wired to have them before the discussion.

Discussion

The fact that LSS prevent inflammation of ECS through autophagy is already known and should be state by the authors on top of their observation.

There are also some issues with the English (verbs conjugation mostly) in the discussion.

Overall, they should better state what is not new and what is new.

We reorganized the discussion and highlighted the suggestions you mentioned in yellow in the revised manuscript.

General questions

Low or oscillatory flow also drive inflammation of the ECs. It is one of the main trigger of AS formation, even before or without any associated inflammatory molecules effect. Does this axis (LOC/miR/EPHB2) down regulated in these pro-atherogenic flow condition? Does upregulating the identified axis would prevent oscillatory-flow-induced inflammation?

Your suggestion is meaningful, thank you very much. Our next research will further explore the relationship between pro-atherosclerotic factors such as oscillatory flow and this axis. However,

we are still purchasing equipment for oscillatory flow, so we are very sorry that we cannot add this part of the experimental verification at this time.

Reviewer #3 (Remarks to the Author):

The authors have investigated the mechanisms by which LSS (12dynes/cm for 12h) promotes anti-inflammatory signalling in cultured human endothelial cells. They demonstrate that LSS reduces TNF α -induced inflammation via increased autophagy. They also deduced from RNAseq data that EPHB2 can be induced by LSS and that this may regulate autophagic flux. Lnc and miRNA that could influence EPHB2 expression were identified and validated in EC exposed to LSS. Evidence is presented suggesting that LOC10798635 acts as a competitive endogenous RNA, reducing the action of miRNA-128-3p and so increasing the expression and activity of EPHB2. This leads to increased HMGB nuclear localisation and regulation of key autophagic components.

This is a very comprehensive study with a lot of data presented and overall I think this is a strong manuscript and contributes to our knowledge of how LSS regulates cytoprotective responses. There are a few issues with the details in the methods/figure legends as outlined below:

The authors should provide more detail on how the cells were obtained and what ethical guidelines were followed. They should also provide information on what passage the cells were used at and what matrix (if any) cells were seeded onto. Please state whether cells were confluent at the onset of flow exposure.

Thank you for your nice comments on our article. The HAEC and HUVEC cell line were obtained from the Department of Pathogenobiology, Basic Medicine College of Jilin University (Changchun, China), these two kinds of cells were authenticated by cell line authentication service in Shanghai Subgene Institute and Shanghai Chuanqiu Biotechnology Co., LTD., respectively, and the identification results showed that the cells were not contaminated by known cell lines and

were single source cell lines.

Adherent cells were digested with trypsin-EDTA digestion solution (0.25%, MRC, USA) and then passaged.

There is a rectangular groove in the middle of the lower chamber for placing slides pretreated with poly-L-lysine solution (Solarbio, China). After the cells on the slides were at 80%–90% confluence, the culture fluid flowed over the surface, and LSS at 12 dyn/cm² was applied to the cells.

We have added the above information and highlighted in yellow in the materials and methods in our revised manuscript.

Relating to cells and independent experiments – when it states ‘three independent experiments’ does this mean that cells were used from three different donors??

I’m very sorry for the misunderstanding. The ‘three independent experiments’ means that an experiment was repeated three times.

The authors should explain the rationale for returning EC to static conditions for 24h after exposure to LSS; it is not clear how this relates to the physiological setting where flow is continuous.

We don’t have much reference for this part of the experimental design. Our initial idea is to ensure that the cells return to steady state growth for a certain length of time after the action of LSS, and to ensure that the amount of cells that restore steady state growth is moderate, they can’t overgrow on the slide. So we chose the 24 hour time point for verification.

Figure 1. Please clarify the MAP1LC3B1 blot in the figure legend – was this used to quantify levels of LC3B2??

Yes, in the process of autophagy, a part of LC3 I is transformed into LC3 II under the action of ATG7 and ATG12-ATG5-ATG16L and binds to the autophagosome membrane, so LC3 II/ I is usually used to measure intracellular autophagy level.

It is not clear from the figure legends or main text how the TNF-treated experimental group was carried out – was TNF added for 12h and then removed and the cultures maintained under static conditions? Was TNF α present throughout the period of flow exposure? The data showing that LSS reduces TNF α -induced inflammation via activation of autophagy is interesting however the physiological relevance is unclear as it seems as though TNF α was applied to static cultures and then LSS was applied, it would be more informative to conduct this experiment where TNF α is added to cells in the presence of flow and assess the impact of autophagy inhibition.

TNF- α was present throughout the period of flow exposure in the experiment of LSS inhibiting the inflammatory response of endothelial cells stimulated by TNF- α (10 ng/ml), and we have highlighted this in yellow in the revised manuscript.

Comment on why only lower doses of lnc increase EPHB2 expression (Fig 4C) whilst all doses affect 128-3p expression.

We found this interesting phenomenon in our experiments. It is not that the higher the expression of LOC107986345, the lower the expression of miR-128-3p. We think that this mechanism of ceRNA in cells interacts with each other in a range of concentrations. When the concentration of LncRNA is very high, the interaction range is exceeded, so the expression of microRNA will not decrease as expected. In this study, when LOC107986345 reached an appropriate concentration (0.001 $\mu\text{g}/\mu\text{L}$), the expression of miR-128-3p decreased the most, and the expression of EPHB2 also increased the most.

The authors should comment on the limitation that only LSS was studied relative to static culture - this is important considering that atherosclerotic lesions develop typically in regions of the vasculature exposed to non-uniform/disturbed flow.

Thank you very much for your suggestion. Our next research is to explore the relationship between LOC107986345/miR-128-3p/EPHB2 axis and disturbed flow. For example, does this axis down regulated in these pro-atherogenic flow condition? Does upregulating the identified axis would prevent disturbed-flow-induced inflammation? However, we are still purchasing disturbed-flow equipment, and we believe we can conduct further research on this research in the near future.

The authors should also comment on the duration of flow exposure and why the 12h timepoint, when EC are still adapting to shear stress, was chosen. It would be beneficial to show the involvement of this pathway under chronic conditions (over 48h)

Regarding the length of time that LSS acts on endothelial cells, we considered that after removing LSS, we need to conduct steady-state culture for 24 hours. In order to ensure the state of the cells throughout the experiment (not overgrowth on the glass slide), we chose LSS to act on endothelial cells 12 hours.

The concluding statement is a bit of a leap with regards to the potential to 'prevent and treat' atherosclerosis considering that this is solely based on in vitro work. The authors should comment on this limitation and comment on future work required to support this statement.

We rewrite the Abstract in the revised manuscript to ensure that it fits this research.

Minor comments

Shear stress should be defined as the frictional force per unit area

Many of the individual figure panels are too small

Figure 2 should be included as a supplementary figure

In the revised manuscript, we make this description: LSS is the frictional force of blood flow acting on the per unit surface of vascular endothelial cells.

We rearranged the Figures in the revised manuscript for better reading and understanding. In the revised manuscript, we deleted the trend analysis of lncRNAs and displayed a more intuitive trend analysis of mRNAs in Fig. 2B.

Reviewers' comments:

Reviewer #1 (Remarks to the Author):

The revised manuscript by Meng et al., has addressed my initial concerns and accurately presented the data. However, the authors have not shown the transfection efficiency of the plasmids used in the study. It would be important to demonstrate that the effect of over-expression or knock-down is not limited to a few cells.

Moreover, the authors reported "Indeed, the transfection efficiency of primary endothelial cells is very low. However, the cells used in this study are cell lines". Are the HAEC and HUVEC used in this study immortalized cells? Because usually HAEC and HUVEC are primary cells and not cell line.

Reviewer #2 (Remarks to the Author):

Reviewing "Laminar shear stress-sensitive long non-coding RNA, LOC10798635, inhibits inflammation in an autophagy-dependent pathway in human aortic endothelial cells via regulation of EPHB2"

Introduction

The introduction has been well improved

Results

Overall figure legend :

There is still discrepancy in the figure description sometime passive form is used, sometime active form. Sometime there is scientific conclusion in the legend (which should not be there). The N for each experiment is not mentioned in the figure legend which does not allow to assess for the robustness of the results.

Figure 1:

I maintain my comment on the use of cell that are not flow sensitive. This is a useless experiment that prove nothing. As those cells are not flow sensitive, they do not carry the receptors and mechanistical macherinery to respond to shear so that appears obvious that they will not display any flow response.

It would have been interesting to show that flow-induced autophagy is endothelial specific by using flow sensitive epithelial cells (as I stated before), first showing that indeed those cells are flow-sensitive and then showing that those cells do not activate autophagy under flow.

I don't ask the authors to do these experiments here, but they can not use non-flow sensitive cell to claim that the flow induce autophagy is endothelial specific.

Figure 3 :

The images of 3d are of too poor quality

"First, the PPI network analysis of EPHB2, HMGB1 and BECN1 showed that there was an interaction between the three of them » : can the author detailed which type of interaction ? data mining ? similarity ? experimental proofs ?

"and we were pleasantly surprised to find that there was an interaction between EPHB2 and HMGB1" : this is not a scientific writing.

Figure 4 :

4e : the author claimed that HMGB1 translocation into the nucleus is increased but the blot presented rather show the opposite. Please clarify.

Page 8 :

"So we want to explore whether EPHB2 was regulated by certain ceRNAs mechanisms under the action of LSS" : problem in verb conjugation for this sentence.

"The corresponding LOC107986345 (NCBI Reference Sequence: XR_001742403.1, Table 6) and

LOC107986196 were verified by qRT-PCR » : why citing the NCBI ref for one and not for the other ?

Figure 5 :

The text is unreadable at printable scale. Please modify. I can not assess the figure l and m because even on the PDF by increasing the size of the figure it is pixelized and not readable. I maintain my 2 comments about the fold change in mir128 level : the fold change for the miR-128-3p, even if significant, is about 10%, how confident the author is that this small decrease in miR-128-3p is responsible for the upregulation of EPHB2 under LSS ? How the important change (more than 90% decrease) in the corresponding lncRNA can translate in such a little effect on miR-128-3p expression ? I agree that the following experiments are confirming the role of mir128 but I am wondering how trustable is the 5d graph and if technical issues happened (that would have minimize the differences).

Figure 6 :

6c : higher magnification of the observed autophagosomes should be provided to be able to assess the double membrane and dense contente.

Mat and Meth

Page 12 : I don't understand the comment on the "cell lines", HUVEC and HAEC are primary cells, not cell lines, what do the others want to state ?

Page 13 : the authors explain that their constructs contained puromycin resistance sequence but they do not say if they used it to purified their population. This should be detailed.

Page 17 : it is unclear if the adhesion of monocyte was done under LSS directly or after cell exposure to LSS.

Discussion

The discussion have been well improved.

Reviewer #3 (Remarks to the Author):

Comments on Revisions

Thank you for addressing my comments, most responses and amendments are satisfactory and have aided in the understanding of the research performed however a couple of issues remain (outlined below). Now that some points have been clarified and I can better understand what was done, I think the article would benefit from a section in the discussion on the limitations of the study, particularly in terms of the cells used and the application of shear stress on an acute time-scale.

Previous comment: The authors should provide more detail on how the cells were obtained and what ethical guidelines were followed. They should also provide information on what passage the cells were used at and what matrix (if any) cells were seeded onto. Please state whether cells were confluent at the onset of flow exposure.

The HAEC and HUVEC cell line were obtained from the Department of Pathogenobiology, Basic Medicine College of Jilin University (Changchun, China), these two kinds of cells were authenticated by cell line authentication service in Shanghai Subgene Institute and Shanghai Chuanqiu Biotechnology Co., LTD., respectively, and the identification results showed that the cells were not contaminated by known cell lines and were single source cell lines. Adherent cells were digested with trypsin-EDTA digestion solution (0.25%, MRC, USA) and then passaged. There is a rectangular groove in the middle of the lower chamber for placing slides pretreated with poly-L-lysine solution (Solarbio, China). After the cells on the slides were at 80%–90% confluence, the

culture fluid flowed over the surface, and LSS at 12 dyn/cm² was applied to the cells. We have added the above information and highlighted in yellow in the materials and methods in our revised manuscript.

Thank you for clarifying. Since HUVEC/HAEC are typically studied as primary cultures it would be helpful to include details describing exactly how these cell lines were generated. Are they immortalised? The current description is insufficient. Please state at what passage these cells were used; this is important to aid in reproducibility of the study. Please also provide evidence that EC have not differentiated and still express EC markers. Please comment on whether the shear responses observed in these cells are normal (compared to primary cultures). You should comment more clearly in the discussion that these experiments were in a cell line and so may not be representative of the physiological state.

Previous comment: The authors should explain the rationale for returning EC to static conditions for 24h after exposure to LSS; it is not clear how this relates to the physiological setting where flow is continuous.

We don't have much reference for this part of the experimental design. Our initial idea is to ensure that the cells return to steady state growth for a certain length of time after the action of LSS, and to ensure that the amount of cells that restore steady state growth is moderate, they can't overgrow on the slide. So we chose the 24 hour time point for verification.

The reasoning is still not clear to me and does not appear to be explained well in the manuscript.

Previous comment: The authors should comment on the limitation that only LSS was studied relative to static culture -this is important considering that atherosclerotic lesions develop typically in regions of the vasculature exposed to non-uniform/disturbed flow.

Thank you very much for your suggestion. Our next research is to explore the relationship between LOC107986345/miR-128-3p/EPHB2 axis and disturbed flow. For example, does this axis down regulated in these pro-atherogenic flow condition? Does upregulating the identified axis would prevent disturbed-flow-induced inflammation? However, we are still purchasing disturbed-flow equipment, and we believe we can conduct further research on this research in the near future.

This should be acknowledged as a limitation in the discussion and/or an area for further research.

Previous comment: The authors should also comment on the duration of flow exposure and why the 12h timepoint, when EC are still adapting to shear stress, was chosen. It would be beneficial to show the involvement of this pathway under chronic conditions (over 48h)

Regarding the length of time that LSS acts on endothelial cells, we considered that after removing LSS, we need to conduct steady-state culture for 24 hours. In order to ensure the state of the cells throughout the experiment (not overgrowth on the glass slide), we chose LSS to act on endothelial cells 12 hours.

The reasoning is still not clear to me and again, I believe this should be acknowledged as a limitation (or caveat to the research) in the discussion. Laminar shear stress induces endothelial quiescence and dramatically reduces proliferation (in primary cultures) and so overgrowth should not be an issue. Do these cell lines that were used continue to proliferate under shear stress? If so this may question their relevance for studying responses to shear stress?? Please comment on this.

Reviewers' comments:

Reviewer #1 (Remarks to the Author):

The revised manuscript by Meng et al., has addressed my initial concerns and accurately presented the data. However, the authors have not shown the transfection efficiency of the plasmids used in the study. It would be important to demonstrated that the effect of over-expression or knock-down is not limited to a few cells.

Thanks for your careful comments, taking the transfection overexpression plasmid pri-EPHB2 (GFP) as an example (see the figure below), the transfection efficiency of the plasmids used in the study is around 90%. So the effect of overexpression or knock-down is not limited to a few cells.

Moreover, the authors reported "Indeed, the transfection efficiency of primary endothelial cells is very low. However, the cells used in this study are cell lines". Are the HAEC and HUVEC used in this study immortalized cells? Because usually HAEC and HUVEC are primary cells and not cell line.

Yes, considering the limited number of primary endothelial cells and the harsh culture conditions, we tried several times in the LSS model using primary endothelial cells, but the results were not satisfactory, coupled with the problem of low transfection efficiency, forcing us to use immortalized HAECs and HUVECs in this study.

Reviewer #2 (Remarks to the Author):

Reviewing “Laminar shear stress-sensitive long non-coding RNA, LOC10798635, inhibits inflammation in an autophagy-dependent pathway in human aortic endothelial cells via regulation of EPHB2”

Introduction

The introduction has been well improved

Results

Overall figure legend :

There is still discrepancy in the figure description sometime passive form is used, sometime active form. Sometime there is scientific conclusion in the legend (which should not be there).

The N for each experiment is not mentioned in the figure legend which does not allow to assess for the robustness of the results.

We have corrected these issues you mentioned and they have been highlighted in yellow in the revised manuscript. The ‘three independent experiments’ in the figure legend means that an experiment was repeated three times.

Figure 1:

I maintain my comment on the use of cell that are not flow sensitive. This is a useless experiment that prove nothing. As those cells are not flow sensitive, they do not carry the receptors and mechanical machinery to respond to shear so that appears obvious that they will not display any flow response.

It would have been interesting to show that flow-induced autophagy is endothelial specific by using flow sensitive epithelial cells (as I stated before), first showing that indeed those cells are flow-sensitive and then showing that those cells do not activate autophagy under flow.

I don't ask the authors to do these experiments here, but they can not use non-flow sensitive cell to claim that the flow induce autophagy is endothelial specific.

Thank you for your persistence on this problem, we also realized that there is indeed something inappropriate. We have removed the part of your concern in the revised manuscript, and your suggestions will be carefully considered in the following research.

Figure 3 :

The images of 3d are of too poor quality

“First, the PPI network analysis of EPHB2, HMGB1 and BECN1 showed that there was an interaction between the three of them » : can the author detailed which type of interaction ? data mining ? similarity ? experimental proofs ?

“and we were pleasantly surprised to find that there was an interaction between EPHB2 and HMGB1” : this is not a scientific writing.

We have improved the 3d image quality in the revised manuscript.

The interaction type of EPHB2 and HMGB1 are textmining, experimentally determined and co-expression, and the interaction type of HMGB1 and BECN1 are textmining, experimentally determined, co-expression and curated databases.

We have corrected the writing and highlighted it in yellow in the revised manuscript.

Figure 4 :

4e : the author claimed that HMGB1 translocation into the nucleus is increased but the blot presented rather show the opposite. Please clarify.

In figure 4e, we want to state that "overexpression of EPHB2 could induce the nuclear translocation of HMGB1 and then interact with EPHB2" rather than "HMGB1 translocation into the nucleus is increased"

Page 8 :

"So we want to explore whether EPHB2 was regulated by certain ceRNAs mechanisms under the action of LSS" : problem in verb conjugation for this sentence.

"The corresponding LOC107986345 (NCBI Reference Sequence: XR_001742403.1, Table 6) and LOC107986196 were verified by qRT-PCR » : why citing the NCBI ref for one and not for the other ?

We changed this sentence to "We want to explore whether EPHB2 was regulated by certain ceRNAs mechanisms when HAECs were under the action of LSS." and highlighted in yellow in the revised manuscript.

Since the study of LOC107986196 was not involved in the following experiments, we did not supplement the relevant information of LOC107986196 in the manuscript.

Figure 5 :

The text is unreadable at printable scale. Please modify. I can not assess the figure l and m because even on the PDF by increasing the size of the figure it is pixelized and not readable.

I maintain my 2 comments about the fold change in mir128 level : the fold change for the miR-128-3p, even if significant, is about 10%, how confident the author is that this small decrease in miR-128-3p is responsible for the upregulation of EPHB2 under LSS ?

How the important change (more than 90% decrease) in the corresponding lncRNA can translate in such a little effect on miR-128-3p expression ?

I agree that the following experiments are confirming the role of mir128 but I am wondering how trustable is the 5d graph and if technical issues happened (that would have minimize the differences).

We have enlarged the text of figure 5l and m in the revised manuscript.

We understand your concerns. However, this part of the experiment was done first in the whole experiment process. When the results were meaningful ($p < 0.001$), we carried out follow-up verification experiments. The follow-up verification results further confirmed the previous, so we

didn't bother with the fold change for the miR-128-3p. If needed, we can revalidate the fold change for the miR-128-3p.

Figure 6 :

6c : higher magnification of the observed autophagosomes should be provided to be able to assess the double membrane and dense content.

Based on your suggestion we have provided the higher magnification of the autophagosomes in Fig.S3 in our revised manuscript.

Mat and Meth

Page 12 : I don't understand the comment on the "cell lines", HUVEC and HAEC are primary cells, not cell lines, what do the others want to state ?

Considering the limited number of primary endothelial cells and the harsh culture conditions, we tried several times in the LSS model using primary endothelial cells, but the results were not satisfactory, coupled with the problem of low transfection efficiency in primary endothelial cells, forcing us to use immortalized HAECs and HUVECs in this study.

Page 13 : the authors explain that their constructs contained puromycin resistance sequence but they do not say if they used it to purified their population. This should be detailed.

Taking the transfection overexpression plasmid pri-EPHB2 (GFP) as an example (see the figure below), the transfection efficiency of the plasmids used in the study is around 90%. So we did not use the puromycin resistance sequence for cell selection.

Page 17 : it is unclear if the adhesion of monocyte was done under LSS directly or after cell exposure to LSS.

We added “the adhesion of monocyte was done after HAECs exposure to LSS” in the Materials and Methods and highlighted in yellow in our revised manuscript.

Discussion

The discussion have been well improved.

Reviewer #3 (Remarks to the Author):

Comments on Revisions

Thank you for addressing my comments, most responses and amendments are satisfactory and have aided in the understanding of the research performed however a couple of issues remain (outlined below). Now that some points have been clarified and I can better understand what was done, I think the article would benefit from a section in the discussion on the limitations of the study, particularly in terms of the cells used and the application of shear stress on an acute time-scale.

Previous comment: The authors should provide more detail on how the cells were obtained and what ethical guidelines were followed. They should also provide information on what passage the cells were used at and what matrix (if any) cells were seeded onto. Please state whether cells were confluent at the onset of flow exposure.

The HAEC and HUVEC cell line were obtained from the Department of Pathogenobiology, Basic Medicine College of Jilin University (Changchun, China), these two kinds of cells were authenticated by cell line authentication service in Shanghai Subgene Institute and Shanghai Chuanqiu Biotechnology Co., LTD., respectively, and the identification results showed that the cells were not contaminated by known cell lines and were single source cell lines. Adherent cells were digested with trypsin-EDTA digestion solution (0.25%, MRC, USA) and then passaged. There is a rectangular groove in the middle of the lower chamber for placing slides pretreated with poly-L-lysine solution (Solarbio, China). After the cells on the slides were at 80%–90% confluence, the culture fluid flowed over the surface, and LSS at 12 dyn/cm² was applied to the cells.

We have added the above information and highlighted in yellow in the materials and methods in our revised manuscript.

Thank you for clarifying. Since HUVEC/HAEC are typically studied as primary cultures it would be helpful to include details describing exactly how these cell lines were generated. Are they immortalised? The current description is insufficient. Please state at what passage these cells were used; this is important to aid in reproducibility of the study. Please also provide evidence that EC have not differentiated and still express EC markers. Please comment on whether the shear responses observed in these cells are normal (compared to primary cultures). You should comment more clearly in the discussion that these experiments were in a cell line and so may not be representative of the physiological state.

Thanks for your kind comment, considering the limited number of primary endothelial cells and the harsh culture conditions, we tried several times in the LSS model using primary endothelial cells, but the results were not satisfactory, coupled with the problem of low transfection efficiency, forcing us to use immortalized HAECs and HUVECs in this study. HAECs and HUVECs were previously purchased (Shanghai Chuan Qiu Biotechnology Co., LTD.) by the research group and stored in the laboratory, and the cells still express vascular endothelial cell markers, such as

VEGFR-2 and VE-cadherin (see the figure below). Compared to primary endothelial cells, immortalized HAECs and HUVECs responded normally to LSS. In addition, we discussed the limitations of this study in the revised manuscript and highlighted in yellow.

Previous comment: The authors should explain the rationale for returning EC to static conditions for 24h after exposure to LSS; it is not clear how this relates to the physiological setting where flow is continuous.

We don't have much reference for this part of the experimental design. Our initial idea is to ensure that the cells return to steady state growth for a certain length of time after the action of LSS, and to ensure that the amount of cells that restore steady state growth is moderate, they can't overgrow on the slide. So we chose the 24 hour time point for verification.

The reasoning is still not clear to me and does not appear to be explained well in the manuscript.

We wanted to observe the changes of intracellular gene expression in vascular endothelial cells before and after the action of LSS to find LSS-sensitive genes, highlighting the important role of LSS in maintaining endothelial cell physiology. In addition, in order to ensure the amounts of cells which restore steady state growth is moderate, we chose the 24 hours time point for verification. We are also trying to set this time length for the first time, hoping to give some useful references to future researchers.

Previous comment: The authors should comment on the limitation that only LSS was studied relative to static culture -this is important considering that atherosclerotic lesions develop typically in regions of the vasculature exposed to non-uniform/disturbed flow.

Thank you very much for your suggestion. Our next research is to explore the relationship between LOC107986345/miR-128-3p/EPHB2 axis and disturbed flow. For example, does this axis down regulated in these pro-atherogenic flow condition? Does upregulating the identified axis would prevent disturbed-flow-induced inflammation? However, we are still purchasing disturbed-flow equipment, and we believe we can conduct further research on this research in the near future.

This should be acknowledged as a limitation in the discussion and/or an area for further research.

Thanks for your suggestion, we have added this part to the discussion section of the revised manuscript and highlighted in yellow.

Previous comment: The authors should also comment on the duration of flow exposure and why the 12h timepoint, when EC are still adapting to shear stress, was chosen. It would be beneficial to show the involvement of this pathway under chronic conditions (over 48h)

Regarding the length of time that LSS acts on endothelial cells, we considered that after removing LSS, we need to conduct steady-state culture for 24 hours. In order to ensure the state of the cells throughout the experiment (not overgrowth on the glass slide), we chose LSS to act on endothelial cells 12 hours.

The reasoning is still not clear to me and again, I believe this should be acknowledged as a limitation (or caveat to the research) in the discussion. Laminar shear stress induces endothelial quiescence and dramatically reduces proliferation (in primary cultures) and so overgrowth should not be an issue. Do these cell lines that were used continue to proliferate under shear stress? If so this may question their relevance for studying responses to shear stress?? Please comment on this.

The suggestion you gave is very meaningful, when we initially designed the experiment, we set several lengths of time for LSS to act, such as 12 hours, 24 hours and 48 hours. The experimental results showed that 12 hours of LSS could activate autophagy and inhibit endothelial cell inflammatory response, so we chose 12 hours as the time length and didn't try other time lengths to conduct the follow-up experiments.